# Timescale correlation of shallow trap states increases electrochemiluminescence efficiency in carbon nitrides

Yanfeng Fang[1,4], Hong Yang[1,4], Yuhua Hou[1], Wang Li[1], Yanfei Shen [2]✉, Songqin Liu[1] & Yuanjian Zhang [1,3]✉

Highly efficient interconversion of different types of energy plays a crucial role in both science and technology. Among them, electrochemiluminescence, an emission of light excited by electrochemical reactions, has drawn attention as a powerful tool for bioassays. Nonetheless, the large differences in timescale among diverse charge-transfer pathways from picoseconds to seconds significantly limit the electrochemiluminescence efficiency and hamper their broad applications. Here, we report a timescale coordination strategy to improve the electrochemiluminescence efficiency of carbon nitrides by engineering shallow electron trap states via Au-N bond functionalization. Quantitative electrochemiluminescence kinetics measurements and theoretic calculations jointly disclose that Au-N bonds endow shallow electron trap states, which coordinate the timescale of the fast electron transfer in the bulk emitter and the slow redox reaction of co-reagent at diffusion layers. The shallow electron trap states ultimately accelerate the rate and kinetics of emissive electron-hole recombination, setting a new cathodic electrochemiluminescence efficiency record of carbon nitrides, and empowering a visual electrochemiluminescence sensor for nitrite ion, a typical environmental contaminant, with superior detection range and limit.

Over billions of years, organisms have evolved to become incredibly efficient in energy conversion. This efficiency is essential for life, as it allows organisms to grow, reproduce, and survive. For the same reason, highly efficient interconversion of different types of energy plays a crucial role in both science and technology. Among them, electrochemiluminescence (ECL), a type of light emission produced by electrochemical reactions in the vicinity of electrodes in solution[1–6], has been successfully commercialized in bioassays for more than 150 clinical biomarkers[7–15]. In general, ECL emitters of high efficiency ($\Phi_{ECL}$) play a central role in developing biosensors with superior sensitivity. Nonetheless, due to the intricate kinetic limitations, $\Phi_{ECL}$ of most ECL emitters in aqueous solutions is essentially low.

To address this challenge, various innovative strategies have been proposed. For instance, accelerating electron transfer at interfaces among electrodes, emitters, and co-reactants via shortened distances[16,17], Schottky junctions[18], nanoconfinement effect[19], or catalytic effects[20] leads to a significantly improved $\Phi_{ECL}$ for a variety of emitters. Suppression of non-radiative relaxation[17,21–28] (e.g., by aggregation/crystallization, host-guest recognition, and ligand-induced assembly), and pre-oxidation/reduction of emitters[29,30] provide alternative routes to booster $\Phi_{ECL}$. In principle, the physical and chemical properties of materials are intrinsically bestowed by interplays not only over different length scales but also at variable time scales. Typical ECL with co-reactants contains mixed multiple charge-transfer

[1]Jiangsu Engineering Research Center for Carbon-Rich Materials and Devices, Jiangsu Province Hi-Tech Key Laboratory for Bio-Medical Research, School of Chemistry and Chemical Engineering, Nanjing 211189, China. [2]Medical School, Southeast University, Nanjing 210009, China. [3]Department of Oncology, Zhongda Hospital, Southeast University, Nanjing 210009, China. [4]These authors contributed equally: Yanfeng Fang, Hong Yang. ✉e-mail: Yanfei.Shen@seu.edu.cn; Yuanjian.Zhang@seu.edu.cn

pathways, including electron transfer in bulk emitters, redox reactions at emitters/co-reactants interface, and electron transition between excited and ground states. Notably, however, there are huge timescale mismatches among them from picoseconds to seconds. From an overall perspective, although challenging, unambiguously revealing ECL kinetics of each process is decisive in extracting the rate-determining step and therefore making a timescale reconcilement of them would open a new methodology to further boost $\Phi_{ECL}$. None-theless, to the best of our knowledge, a complete quantitative description and coordination of ECL kinetics for diverse charge transfer processes at different timescales have still been lacking.

Herein, we report a timescale coordination strategy to improve $\Phi_{ECL}$ of carbon nitrides (CN) by engineering shallow electron trap states via Au-N bond functionalization ($Au_x$-CN). For this purpose, a quantitative description of the complete charge transfer kinetics during the ECL of CN from the timescale of picosecond to second was developed, using operando electrochemical impedance spectroscopy (EIS), fs-transient absorption spectroscopy (TAS), transit open circuit photovoltage (OCP) and density functional theory (DFT) calculations. It was revealed the Au-N bonding activated new shallow electron trap states, which worked as

an electron sink in coordinating the timescale differences between the slow redox reaction at diffusion layers and fast electron transfer in bulk CN. Accordingly, more excited electrons and holes were produced, leading to a faster electron transition kinetics between excited and ground states. As a result, the timescale coordination strategy showed a four-fold enhancement of $\Phi_{ECL}$ for CN, setting a new cathodic $\Phi_{ECL}$ record in aqueous solution and co-reagent pathway.

## Results

### Synthesis and structural characterization of $Au_x$-CN photoelectrode

As a metal-free polymeric semiconductor, 2D carbon nitride (CN) has drawn increasing attention as a new generation of conjugated polymer-based ECL luminophore[18,31–33]. It demonstrates intriguing properties, such as abundant availability, high stability, excellent bio-compatibility, and record-level cathodic $\Phi_{ECL}$ among metal-free ECL emitters[34,35]. Herein we take the $Au_x$-CN as a model system to modulate the timescales of each ECL process of CN. The general procedures for CN and $Au_x$-CN photoelectrode preparation were shown in Fig. 1a. Briefly, a clean fluorine doped tin oxide (FTO) glass was immersed into

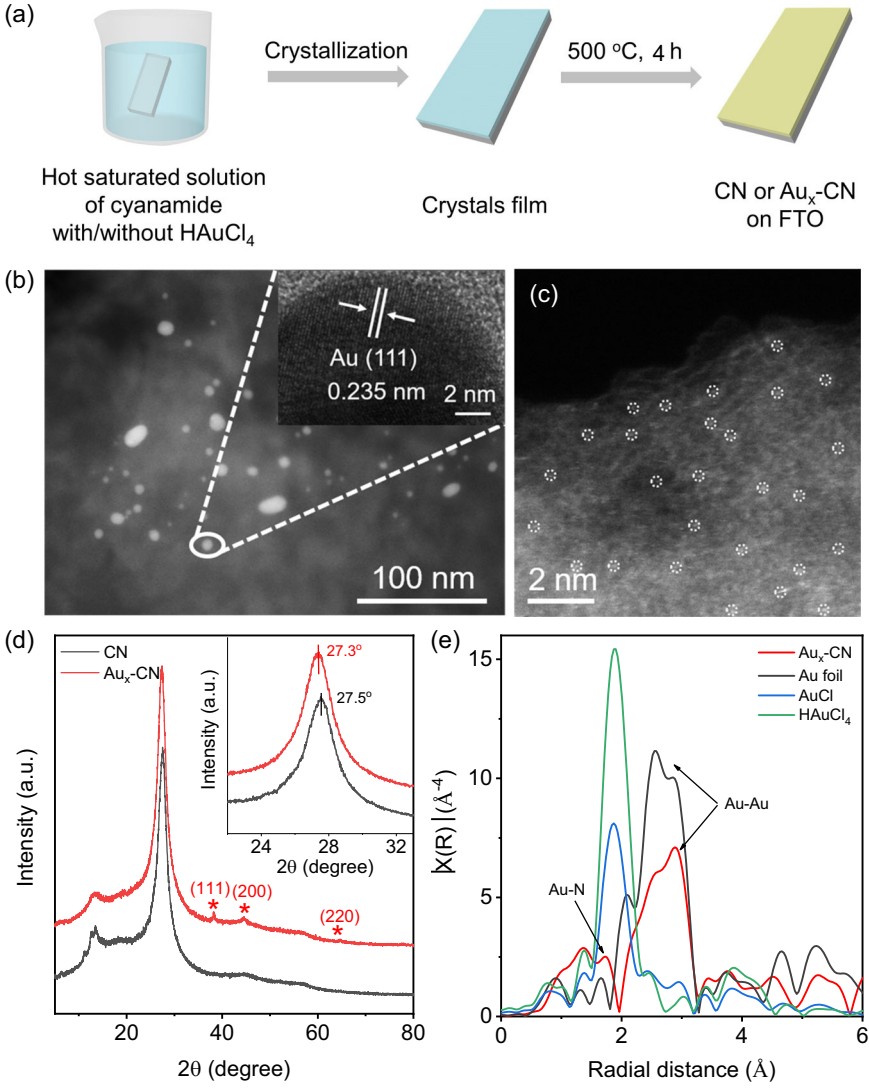

**Fig. 1 | Synthesis and structural characterization of $Au_x$-CN photoelectrode.**
**a** Scheme of the general fabrication procedure for CN and $Au_x$-CN on the FTO.
**b** STEM and **c** HAADF-STEM images of $Au_x$-CN. Inset of **b**: HRTEM image of $Au_x$-CN. Isolated bright spots highlighted by a white circle: Au single atom. **d** XRD patterns

of CN and $Au_x$-CN. The asterisk indicates the diffraction from Au nanoparticles. "a.u." refers to arbitrary units. **e** Fourier transforms of the EXAFS spectra of $Au_x$-CN and the reference samples.

a hot saturated cyanamide solution (80 °C) with or without HAuCl$_4$. A uniform crystal film was firmly coated on FTO after cooling. The final CN and Au$_x$-CN photoelectrodes were prepared by calcination of the crystal films at 500 °C for 4 h under N$_2$ atmosphere. The scanning electron microscopy (SEM) of CN and Au$_x$-CN photoelectrodes exhibited a continuous film on FTO with an intimate interface (Supplementary Fig. 1). Scratch-track morphologies showed no obvious cracks, indicating the good toughness of the CN and Au$_x$-CN photoelectrodes (Supplementary Fig. 2). To disclose the Au species in the interlayer of CN, the scanning transmission electron microscopy (STEM) images were measured. As shown in Figs. 1b and 1c, abundant bright spots were observed, corroborating the co-existence of uniformly dispersed Au nanoparticles (NPs) and single atoms in the adjacent layer of the CN matrix. High-resolution TEM (HRTEM) of Au$_x$-CN showed a typical Au (111) plane with a characteristic lattice spacing of 0.235 nm was observed (Fig. 1b inset)[18,36]. The corresponding high-resolution STEM-energy dispersive spectrometer (EDS) elemental mapping images showed that the C, N, and Au were homogeneously dispersed across the entire CN (Supplementary Fig. 3). The inductively coupled plasma-optical emission spectroscopy (ICP-OES) analysis showed that the Au loading in Au$_x$-CN was 0.06 wt %, indicating only a minor structural disorder was introduced into the pristine CN structure.

The layered crystal structures of Au$_x$-CN were investigated by X-ray diffraction (XRD, Fig. 1d). A series of characteristic diffraction peaks at $2\theta = 34.4°$, 44.7° and 64.8° belonging to (111), (200) and (220) crystal planes of Au NPs were slightly observed for Au$_x$-CN, further confirming the cubic crystal structure of the intercalated Au NPs[37]. Moreover, the XRD peak of Au$_x$-CN at 27.3° was observed, which could be assigned to interlayer stacking (002) of the conjugated aromatic systems[38]. Compared to that of CN (27.5°), it was shifted to a smaller angle, which was consistent with TEM analysis, indicative of a thicker interlayer distance due to intercalated Au species[39]. The Fourier transform infrared spectroscopy (FTIR, Supplementary Fig. 4), X-ray photoelectron spectroscopy (XPS, Supplementary Fig. 5) and matrix-free laser desorption/ionization time-of-flight mass (LDI-TOF-MS, Supplementary Fig. 6) spectra verified the intercalated Au rarely damaged the CN framework. The changes in the energy band structure were due to the incorporation of the Au species into CN (Supplementary Fig. 7).

To explore the chemical state of Au in Au$_x$-CN, the X-ray absorption near-edge structure (XANES) spectroscopy was conducted at the Au L3-edge. As shown in Supplementary Fig. 8, the absorption edge for Au$_x$-CN was located between HAuCl$_4$ and Au foil references, and closer to that of AuCl, implying that the Au species carried slightly positive charges[40,41]. The valance state of Au$_x$-CN was further confirmed by the XPS spectra. Supplementary Fig. 9 showed the typical $4f^{5/2}$ and $4f^{7/2}$ signals at 89.7 and 83.4 eV, assigned to the oxidation state (Au$^I$) and metallic state (Au$^0$) in Au$_x$-CN[40], respectively. To disclose the coordination environment of Au species in Au$_x$-CN, the extended X-ray absorption fine structure (EXAFS) spectroscopy was investigated at the Au L3-edge. Figure 1e shows the Fourier transforms of the Au L3-edge EXAFS oscillations of the as-prepared Au$_x$-CN. It was obvious that the peaks at 2.0-3.3 Å were assigned to the Au-Au bond corresponding to Au NPs, while the peak located at approximately 1.7 Å could be ascribed to the scattering path of Au-N(C)[40,41]. According to this bond length, the density functional theory (DFT) calculation in the following discussion further verified that the Au-N bond was positioned between the interlayer of CN, rather than the cavity of the tri-s-triazine framework in the basal plane of CN. Therefore, by a simple co-polymerization method, Au-N bonding was introduced into the interlayer of CN.

## ECL performance of Au$_x$-CN photoelectrode

As shown in Fig. 2a, a minor current in cyclic voltammogram (CV) curves were observed for both CN and Au$_x$-CN photoelectrode in the solution without K$_2$S$_2$O$_8$, indicating negligible polarization of water during the reduction of K$_2$S$_2$O$_8$. Notably, the reduction peak of K$_2$S$_2$O$_8$ was out of the scope of the electrochemical window (Supplementary Fig. 10). It was attributed to the high iR drop of CN and Au$_x$-CN photoelectrode[42]. Figure 2b showed the ECL onset potential of Au$_x$-CN photoelectrode positively shifted by 200 mV compared to that of CN photoelectrode. As shown in Supplementary Fig. 11, different Au loadings were examined to assess the ECL intensity of the Au$_x$-CN photoelectrode, and the intensity reached almost 4 times that of the CN photoelectrode. The ECL of Au$_x$-CN photoelectrode was stable under continuous CV scans (Fig. 2c). The intense emission can be easily observed by the naked eyes and the uneven luminescence on the photoelectrodes may be attributed to differences in surface flatness after the thermal condensation (Fig. 2d). Figure 2e showed the ECL peak centered at ca. 455 nm, almost identical to the fluorescence (FL) spectrum, manifesting the CB-VB transition mechanism, different from many nanostructured ECL emitters with defective state emission[35,43]. An easily reproducible Ru(bpy)$_3$$^{2+}$/K$_2$S$_2$O$_8$ aqueous system was used as a reference to facilely compare $\Phi_{ECL}$ of different luminophores in this study (see the detailed discussion of the justification and calculation methods in the Experimental section of Supplementary Information and Supplementary Fig. 12 and 13). As shown in Fig. 2f, the Au$_x$-CN photoelectrode reached 3261 times the aqueous Ru(bpy)$_3$$^{2+}$/K$_2$S$_2$O$_8$ reference, which set a record of $\Phi_{ECL}$ for the CN family and was higher than those of most reported luminophores, to the best of our knowledge (Supplementary Table 1).

Electron transfer processes of ECL in Au$_x$-CN. The co-reactant typed ECL (e.g., Ru(bpy)$_3$$^{2+}$) has been elucidated in detail for four possible reaction routes (Supplementary Fig. 14)[44,45]. Three of them are involved in the direct oxidation of co-reactants on the electrode surface (Supplementary Fig. 14a-c). Especially, for the CN and Au$_x$-CN photoelectrode, the redox reaction was evidently inhibited by ca. 1000 times compared with bare FTO in regard to the interfacial charge transfer resistance across the electrode/electrolyte (R$_{ct}$, Supplementary Fig. 15 and Supplementary Table 2). It suggested that the co-reactants accepted electrons from CN or Au$_x$-CN, rather than the conventional substrate electrode. The general ECL mechanisms of Au$_x$-CN/K$_2$S$_2$O$_8$ system are shown in Eqs. (1)–(4)[35]. Briefly, electrons were injected from FTO substrate electrodes into the conduction band (CB) of CN to form Au$_x$-CN$^{•−}$ (Eq. (1)). Next, a few excited electrons obtained from Au$_x$-CN$^{•−}$ reduced co-reagents, producing strong oxidant SO$_4$$^{•−}$ (Eq. (2)), which subsequently generated holes in the valence band (VB, Eq. (3)) by an additional one-electron extraction. Lastly, the electrons in the CB and the holes in the VB recombined with the emission of light (Eq. (4)). The schematic illustration of charge transfer during ECL processes is shown in Fig. 3a.

$$Au_x\text{-}CN + e^- \rightarrow Au_x\text{-}CN^{•-} \tag{1}$$

$$S_2O_8^{2-} + Au_x\text{-}CN^{•-} \rightarrow SO_4^{2-} + SO_4^{•-} + Au_x\text{-}CN \tag{2}$$

$$Au_x\text{-}CN^{•-} + SO_4^{•-} \rightarrow Au_x\text{-}CN^* + SO_4^{2-} \tag{3}$$

$$(SO_4^{•-} \rightarrow SO_4^{2-} + h^+)$$
$$Au_x\text{-}CN^* \rightarrow Au_x\text{-}CN + h\nu \tag{4}$$

## Electron transfer kinetics in bulk ECL emitters

To verify electron transfer in the bulk ECL emitter (Eq. 1), the EIS under different potentials, the fs-TAS, and the OCP were measured. Before that, the photothermal effect and the plasma resonance effect were excluded by control experiments, as negligible temperature variations

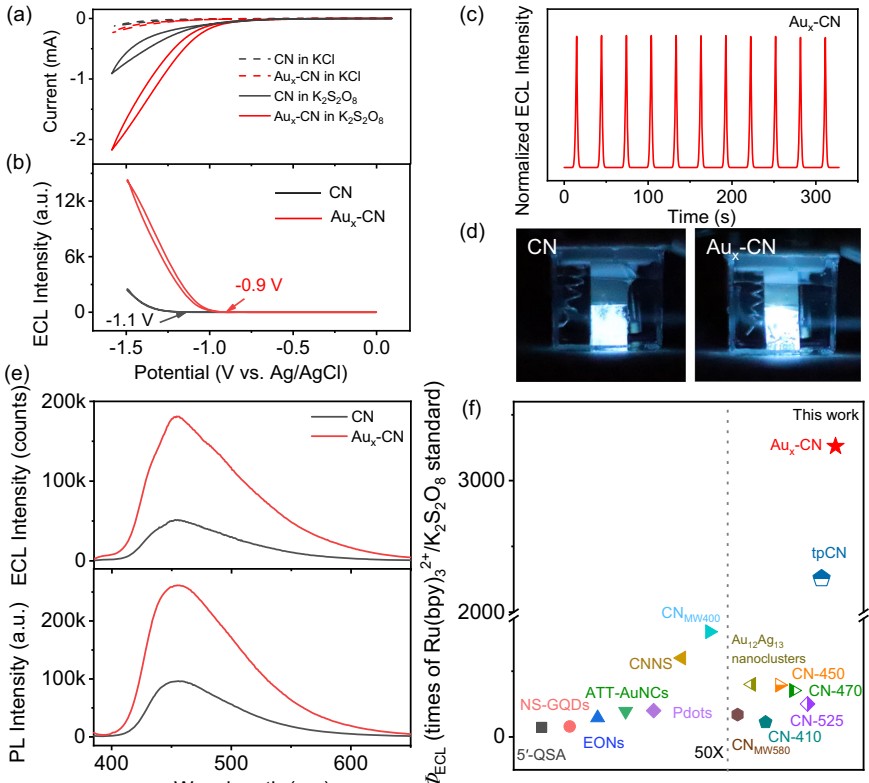

**Fig. 2 | ECL performance of Au_x-CN photoelectrode. a** CV and **b** ECL curves of CN and Au_x-CN photoelectrode. Electrolytes: 0.01 M phosphate buffer saline containing 0.1 M KCl with or without 25 mM $K_2S_2O_8$. "a.u." refers to arbitrary units. **c** ECL response of CN and Au_x-CN photoelectrode under continuous CV scans. **d** Photographs of CN and Au_x-CN photoelectrode under -1.5 V vs. Ag/AgCl. **e** ECL and fluorescence spectrum of CN and Au_x-CN photoelectrode. "a.u." refers to

arbitrary units. **f** Relative ECL efficiency comparison of Au_x-CN photoelectrode and other counterparts in previous reports: 5'-QSA[78], NS-GQDs[65], EONs[79], ATT-AuNCs[21], CNNS[80], Pdots[16], CN_MW400[35], CN_MW580[42], Au12Ag13 nanocluster[81], CN-410[31], CN-450[31], CN-470[31], CN-525[31], tpCN[34]. The left panel divided by the dashed line shows magnified data (50x) and the data points to the left and right-hand side of the dashed line follow the same y scales.

(Supplementary Fig. 16) and plasma resonance effects of Au species (Supplementary Fig. 7b) were observed for the CN and Au_x-CN photoelectrodes. As shown in Fig. 3b, c, and Supplementary Fig. 17, at the low voltage, Nyquist plots revealed approximated straight lines, which indicated high charge reaction resistance. When the voltage increased, a more complete semicircle appeared in the Nyquist plots, suggesting the Faradaic reaction occurred[46,47]. Moreover, the CN and Au_x-CN photoelectrode started to exhibit Warburg impedance at −1.5 V and −1.0 V, respectively. The lower voltage value indicated that the CN photoelectrode had a limit in charge transfer resistance, presumably due to its poor electronic conductivity[46,47]. As shown in Fig. 3d, the Nyquist plots showed straight lines in the high-frequency part at −1.5 V. This type of EIS pattern belonged to the transmission line model (Supplementary Fig. 18)[48,49], and the simplified equivalent circuit models were shown in Fig. 3d inset (see the fit parameters in Supplementary Tables 3 and 4). As summarized in Supplementary Fig. 19, the values of electron transport resistance (R_t) remained almost constant under different applied voltages, indicating R_t was an intrinsic property of emitters. The R_t values for CN photoelectrode were approximately 5 times higher than that of Au_x-CN photoelectrode, suggesting the improved electron conductivity of Au_x-CN photoelectrode[47].

To confirm the carrier diffusion dynamics in bulk emitters, the carrier diffusion lifetime (τ_d) and electron mobility (μ) of CN and Au_x-CN photoelectrode at ECL work conditions were also analyzed from the EIS spectra. As shown in Fig. 3d, the inflection point between the straight line and arc in the high-frequency part was associated with τ_d, which was inversely correlated with the frequency[50,51]. The τ_d of CN and Au_x-CN photoelectrode were calculated to be 99 and 39 μs, respectively. And the electron mobility was calculated using the Nernst-

Einstein equation[51,52]:

$$\mu = \frac{eL^2}{k_B T \tau_d} \quad (5)$$

where e is the electronic charge, L is the effective travel distance of carriers through the active layer (3 μm in this work), k_B is the Boltzmann constant, and T is the absolute temperature. The electron mobility of the Au_x-CN photoelectrode was calculated as $8.98 \times 10^{-2}$ cm² V⁻¹ s⁻¹, which was 3 times higher than that of the CN photoelectrode ($3.54 \times 10^{-2}$ cm² V⁻¹ s⁻¹). The significantly reduced τ_d and improved electron mobility of the Au_x-CN photoelectrode indicated the faster carrier diffusion kinetics in the bulk Au_x-CN photoelectrode. Therefore, in the bulk CN, electron diffusion kinetics was boosted after the introduction of the Au-N bond in the interlayer of CN.

To investigate the electron states in bulk emitters, the fs-TAS was conducted. Figure 3e exhibited a positive absorption from 500 to 780 nm and a negative absorption from 425 to 500 nm in the fs-TAS of CN and Au_x-CN photoelectrode at 5 fs delays under a 365 nm pump. The positive absorption features between 500 and 780 nm, which was highly related to photogenerated electrons in the CN[53,54]. The photoinduced absorption signal in the visible regions was partially quenched for Au_x-CN, indicative of an effective electron transfer between CN and Au species by Au-N bond[54]. To avoid the effects of excitation and emission, the kinetic courses of transient signals at 750 nm were used to study the electron transfer kinetics of CN and Au_x-CN photoelectrode[55]. The fitting results showed that the electron transfer lifetime (τ_ave-shallow) of CN and Au_x-CN photoelectrode were 185.48 and 348.99 ps, respectively (Fig. 3f). The quenched TAS intensity and

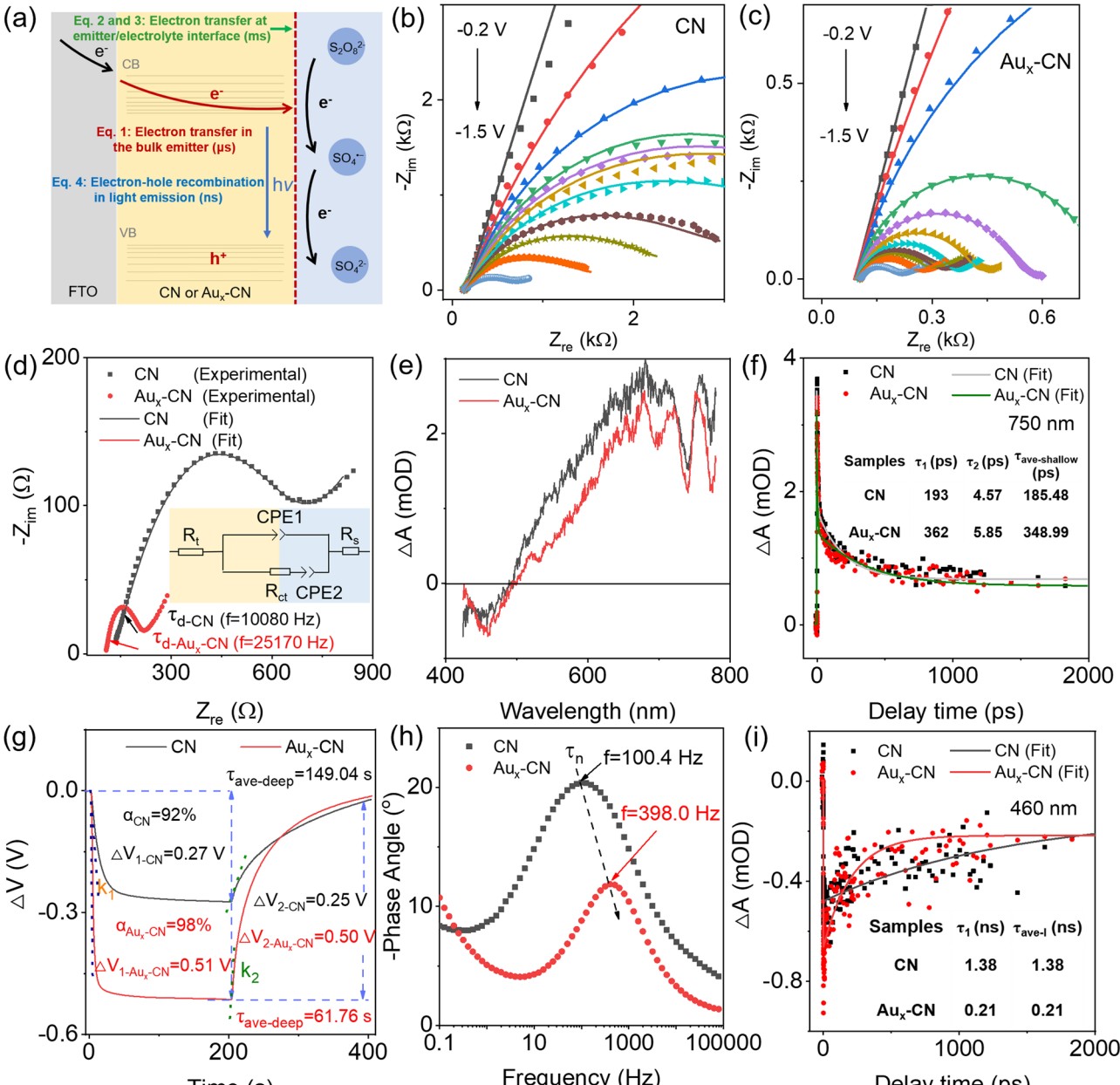

**Fig. 3 | Electron transfer pathways and kinetics of Au$_x$-CN. a** Possible charge transfer processes of ECL in Au$_x$-CN. Nyquist plots for **b** CN and **c** Au$_x$-CN photoelectrode at different applied potentials versus Ag/AgCl in 0.01 M phosphate buffer saline containing 0.1 M KCl and 25 mM K$_2$S$_2$O$_8$. Scatters and lines represent the experiment and fitted data, respectively. Black line: −0.2 V; Red line: -0.4 V; Blue line: −0.6 V; Green line: −0.8 V; Violet line: −0.9 V; Yellow line: −1.0 V; Cyan line: −1.1 V; Wine line: −1.2 V; Dark yellow line: −1.3 V; Orange line: −1.4 V; Light blue line: −1.5 V. **d** Nyquist plots for CN and Au$_x$-CN photoelectrode at −1.5 V vs. Ag/AgCl. Inset: Simplified equivalent circuit. R$_t$: electron transport resistance. R$_{ct}$: the charge reaction resistance. Constant phase element (CPE1): non-ideal capacitance. CPE2: non-ideal Warburg element. R$_s$: solution resistance. **e** Femtosecond transient absorption spectra of CN and Au$_x$-CN photoelectrode at 5 fs delays under a 365 nm pump. **f** Representative ultrafast transient absorption kinetics probed at 750 nm (pump at 365 nm) for CN and Au$_x$-CN photoelectrode. **g** Open circuit potential of CN and Au$_x$-CN photoelectrode under chopped visible light irradiation in 0.1 M KCl. ΔV$_1$ (arrow range): open circuit potential value after light on; ΔV$_2$ (arrow range): the recovery value of photovoltage after light off; k$_1$ (dashed line): the slopes of photovoltage drop after light on; k$_2$ (dashed line): the slopes of photovoltage drop after light off. **h** Bode plots for CN and Au$_x$-CN photoelectrode at −1.5 V vs. Ag/AgCl. **i** Representative ultrafast transient absorption kinetics probed at 460 nm (pump at 365 nm) for CN and Au$_x$-CN photoelectrode.

increased lifetime suggested that the near band-edge shallow electron trap states of CN, most presumably introduced/strengthened by Au-N bonding, offered more opportunities for excited electrons to participate ECL[55].

The OCP under chopped light was further measured to study the deep trapped long-lived electrons state (Fig. 3g). The surface deep electron trap state could be evaluated by the ratio of photovoltage value ($\alpha$) after light on (ΔV$_1$) and off (ΔV$_2$). It is commonly believed that

low $\alpha$ is related to the electron deep trap state[42]. It was observed that the value of $\alpha$ was closer to 1 for Au$_x$-CN photoelectrode, suggesting a less surface-deep electron trap state. The average charge lifetime ($\tau_{ave-deep}$) was calculated by fitting the photovoltage decay curves[56,57]. As shown in Supplementary Fig. 20, the $\tau_{ave-deep}$ for CN photoelectrode was 149.04 s, which was 2 times of magnitude longer than Au$_x$-CN photoelectrode (61.76 s). In general, the long $\tau_{ave-deep}$ indicated severe electron trap effect[57]. Thus, the OCP measurement disclosed that

$Au_x$-CN photoelectrode had less surface deep trapping state of electrons, which would lead to a higher efficiency of electron utilization in ECL. Moreover, the OCP measurement disclosed that $Au_x$-CN photoelectrode had higher excited electron-storage capacity, which was favorable for excitation and faster electron-hole recombination kinetics (see more detailed discussion in Supplementary Fig. 21).

## Electron transfer kinetics at emitter/co-reagent interface

The phase angle vs. frequency plots at different potentials (Supplementary Fig. 22) of CN and $Au_x$-CN photoelectrode were used to probe the co-reagent reduction kinetics at the emitter/$S_2O_8^{2-}$ interface during the ECL process (Eqs. (2) and (3)). The peak at the mediate frequency region is related to the effective lifetime[50,51] ($\tau_n$, Fig. 3h). The $\tau_n$ of CN and $Au_x$-CN photoelectrode from EIS bode plots was approximately 4 times smaller than that of CN photoelectrode at different overpotentials (Supplementary Fig. 23) and the values of $\tau_n$ for CN and $Au_x$-CN photoelectrode was 9.9 ms and 2.5 ms at work conditions. In general, a shorter lifetime was associated with the faster electron transfer kinetic at the emitter/co-reagent interface. As shown in Supplementary Fig. 24, the $R_{ct}$ for $Au_x$-CN photoelectrode obtained from EIS fitting was approximately 10 times smaller than that of CN photoelectrode at different overpotentials (Supplementary Fig. 24), indicating the reduction of $S_2O_8^{2-}$ was more ready to occur at the $Au_x$-CN/$S_2O_8^{2-}$ interface. Therefore, in the bulk CN, electron transfer kinetics at the emitter/co-reagent interface was boosted after the introduction of more efficient shallow electron trap states.

Electron-hole recombination kinetics in light emission. To understand the influencing factors for the emissive state in ECL (Eq. (4)), the fs-TAS, electron-hole recombination efficiency ($\eta_{re}$) and time-resolved FL spectra were measured. Figure 3e exhibited a negative absorption from 425 to 500 nm in the fs-TAS of CN and $Au_x$-CN photoelectrode, which was associated with the stimulated emission[54,56]. The lower TAS intensity indicated increased electron-hole recombination for $Au_x$-CN photoelectrode, which was consistent with the higher FL intensity in Fig. 2e. Accordingly, as shown in Supplementary Fig. 25, the calculated $\eta_{re}$ (Eq. 6) of CN and $Au_x$-CN photoelectrode was calculated to be 47.0% and 65.6%. Moreover, the TA signal of $Au_x$-CN photoelectrode decreased to almost zero within 1 ns after photoexcitation, while that of CN did not, suggesting a fast electron-hole recombination (Supplementary Fig. 26)[56]. The fitting result of the negative signal was demonstrated in Fig. 3i. The electron-hole recombination lifetime ($\tau_l$) of $Au_x$-CN photoelectrode fitted at 460 nm was 0.21 ns and this value was nearly 7 times shorter than that of CN photoelectrode (1.38 ns). Moreover, the FL decay lifetime for CN and $Au_x$-CN photoelectrode was also measured to be 3.46 ns and 1.69 ns, respectively (Supplementary Fig. 27). In general, a shorter TAS and FL decay lifetime of the radiative process often indicates a faster recombination rate for electron-hole pairs[42]. In this sense, the enhanced electron-hole recombination efficiency and decreased FL lifetime were observed, indicative of the critical role of the Au-N bond, most presumably via a more efficient shallow electron trap states.

$$\eta_{re}(\%) = \left(1 - \frac{J_{KCl}}{J_{TEOA}}\right) \times 100\% \qquad (6)$$

$J_{KCl}$ is the photocurrent density obtained in 0.1 M KCl aqueous solution, while $J_{TEOA}$ is the photocurrent density obtained in 0.1 M KCl containing 10% (v/v) triethanolamine (TEOA).

Given the above, the quantitative values of kinetic parameters for electron transfer in the ECL of CN and $Au_x$-CN are shown in Table 1. The roles of Au in the electron transfer in the ECL of $Au_x$-CN were summarized as follows. The formation of Au-N bonds accelerated the electron transfer in the bulk $Au_x$-CN photoelectrode. More importantly, it endowed new shallow electron trap states that had a longer lifetime, thus smoothing out the timescale inconsistencies of electron transfer at different stages. As shown in Table 1 and Supplementary

Fig. 28, the shallow-trapped electrons in the bulk emitter that existed at about the picosecond time scale ($\tau_{shallow}$) extended 2 times magnitude by the Au-N bond functionalization, which coordinated the slow charge transfer at the emitter/co-reactant interface in the millisecond time scale ($\tau_d$) and fast electron transfer in bulk emitter in the microsecond time scale ($\tau_n$). It would increase the reduction of co-reactants at the interface, and further improve the electron-hole recombination rate and efficiency for the ECL of $Au_x$-CN in the nanosecond time scale ($\tau_l$). In contrast, the accumulation of long-lived, deeply trapped electrons in the second time scale ($\tau_{deep}$), detrimental to efficient ECL, was effectively suppressed. It should be noted that while the intricate interplay of charge transfer kinetics in ECL has rarely been quantitatively dissected, the profound influence of charge carrier traps on ECL performance, despite their extensive exploration in organic semiconductors for diverse time scales, remains largely elusive. Harnessing quantitative kinetic study tools like operando EIS, transit OCP, and TAS, this study unveils how incorporating appropriate trap states can manipulate the timescales of substantial fluctuations in electron transfer within bulk ECL emitters, redox reactions at diffusion layers, and electron transitions between excited and ground states, thereby paving the way to enhance ECL efficiency.

## Molecular insights of ECL enhancement by DFT calculations

The influence of Au-N bonds on accelerating the electron transfer and generating new shallow electron trap states in the $Au_x$-CN photoelectrode was explored by the DFT calculation. In a simplified $Au_x$-CN model (see Supplementary Fig. 29–34, see more discussion in Supplementary Information), the Au-N bonds, i.e., the distance between the Au atoms and the two adjacent CN layers, were 2.06 Å and 2.07 Å, respectively, matching well with the EXAFS results (Fig. 1e). The density of states (DOS) and the visible crystal orbitals of pristine CN and $Au_x$-CN were analyzed (Fig. 4). The valence band maximum (VBM) of pristine CN (Fig. 4a and c) mainly originated from the N 2$p$ orbitals; whereas the conduction band minimum (CBM) was contributed both by the C and N 2$p$ orbitals. A similar electronic structure of CBM was found in $Au_x$-CN (Fig. 4b and d), and the contribution from Au 5$d$ and 6$s$ orbitals was negligible; while the VBM was mostly formed by Au 5$d$ and 6$s$ orbitals and N 2$p$ orbitals. As a result, the higher-energy Au 5$d$ and 6$s$ orbitals recombined with the pristine CN π bonding orbitals and π anti-bonding orbitals, which made the continuity of the energy band near the CBM and VBM in $Au_x$-CN be improved. Consequently, charge carriers could more readily transfer in the VBM and CBM, facilitating intralayer charge transfer.

In addition, inter-layer charge transfer was further analyzed by the Bader charge. The Bader charge difference between each adjacent layer of pristine CN was marginal (Supplementary Fig. 35b, |$\Delta q$| roughly

**Table 1 | Quantitative kinetic parameters of electron transfer in the ECL of CN and $Au_x$-CN**

| Electron transfer stage | Parameters | CN | $Au_x$-CN |
|---|---|---|---|
| In bulk ECL emitter | $\tau_d$ (μs)[a] | 99 | 39 |
| | $\mu$ (cm$^2$ V$^{-1}$ s$^{-1}$)[b] | $3.54 \times 10^{-2}$ | $8.98 \times 10^{-2}$ |
| | $\tau_{ave\text{-}shallow}$ (ps)[c] | 185.48 | 348.99 |
| | $\tau_{ave\text{-}deep}$ (s)[d] | 149.04 | 61.76 |
| At the emitter/co-reactant interface | $\tau_n$ (ms)[e] | 9.9 | 2.5 |
| During light emission | $\tau_l$ (ns)[f] | 1.38 | 0.21 |
| | $\eta_{re}$ (%)[g] | 47.0% | 65.6% |

[a]carrier diffusion lifetime; [b]electron mobility; [c]shallow electron trap state lifetime; [d]deep electron trap state lifetime; [e]effective lifetime; [f]electron-hole recombination lifetime; [g]electron-hole recombination efficiency.

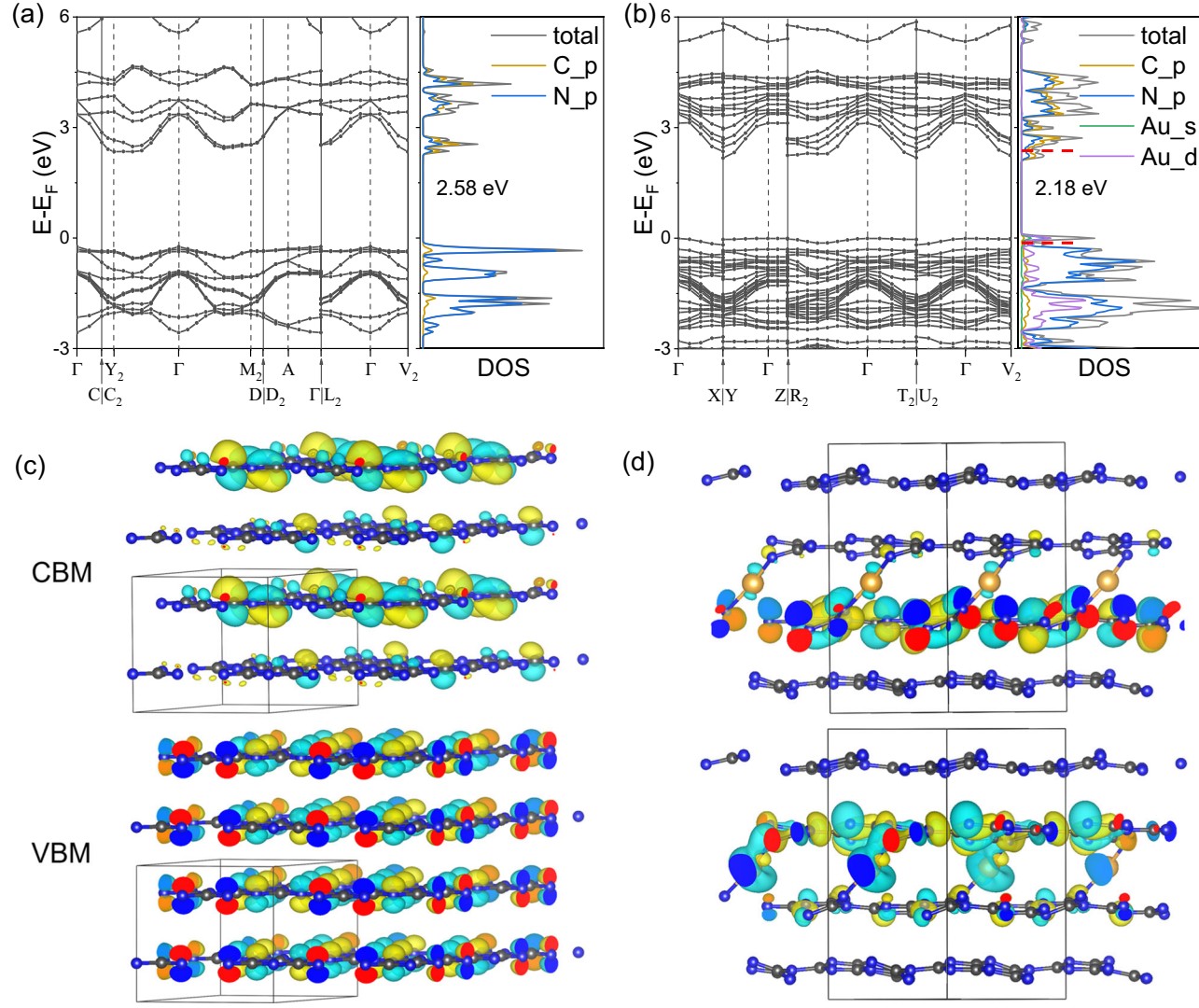

**Fig. 4 | Band structure, DOS, and VBM/CBM crystal orbitals of CN and Au$_x$-CN.** Band structure and DOS of **a** CN and **b** Au$_x$-CN. Red dash lines in Au$_x$-CN mark the positions of CMB and VBM for CN. VBM/CBM crystal orbitals of **c** CN and **d** Au$_x$-CN. Gray coloring indicates carbon atoms, blue indicates nitrogen, and orange indicates gold. Source data are provided as a Source Data file.

0.0009 e⁻), indicative of a very weak inter-layer charge transfer. In contrast, more electrons accumulated between each layer of Au$_x$-CN (Supplementary Fig. 35d, approximately 0.2 e⁻ of layer charge), and Au species lost 0.48 e⁻. As a result, Au$_x$-CN exhibited a much higher interlay charge difference (Supplementary Fig. 35d, |$\Delta q$| roughly 0.142 e⁻ and 0.203 e⁻), indicating Au-N$_x$ in the CN inter-layer served as a bridge, promoting the electron transfer between the adjacent layers. Thus, the Au-N bonding improved the overall electron transfer in Au$_x$-CN, compared to the pristine CN.

Considering that the Au content was only 0.06 wt% in the as-prepared Au$_x$-CN, the band structure in Au$_x$-CN should still be dominated by the pristine CN. Accordingly, the band gap of Au$_x$-CN was 2.79 eV, which was only 0.03 eV lower than the pristine CN (Supplementary Fig. 7). Nonetheless, the large-nuclear-charge Au species enhanced conjugation within the CN layer and made CBM shift down by 0.2 eV (red dash line in Fig. 4b). This new state close to the CBM energy level position of pristine CN, was usually considered to be shallow electron trap states, in which captured electrons could be released back to the band under external stimuli, such as an electric field, thereby reconciling the timescale difference of each step in ECL[58-60]. Therefore, the Au-N bonds not only accelerated the electron transfer in the bulk CN, but also created new shallow electron trap states near the CBM, two important factors in the improved ECL efficiency for CN.

## Enhanced sensitivity in NO$_2^-$ detection using Au$_x$-CN

The nitrite ion (NO$_2^-$), as a notorious pollutant, has gained significant interest due to its widespread presence in drinking water and various food products[61,62]. This prevalence has spurred intense research into the NO$_2^-$ detecting. In general, NO$_2^-$ could consume the SO$_4^{·-}$ around the electrode surface, resulting in a decrease of ECL intensity (Fig. 5a)[42]. Figure 5b and 5c showed that the cathodic ECL signal decreased gradually with the increase of NO$_2^-$ concentration for both Au$_x$-CN and CN nanosheets photoelectrode. The logarithmic value of the ECL intensity at the Au$_x$-CN photoelectrode scaled linearly with the concentration of NO$_2^-$ from $1 \times 10^{-15}$ to $1 \times 10^{-9}$ M, with a very low detection limit of 0.21 fM. The excellent performance of the NO$_2^-$-based ECL biosensor has made it one of the most sensitive signal-amplification-free biosensors (Supplementary Table 5). As shown in Supplementary Fig. 36, the ECL intensity of the biosensor for detection of NO$_2^-$ had little change under continuous potential scanning for more than 10 cycles, which indicated the good stability of the biosensor. And, the slope of the calibration curve and linear range of Au$_x$-CN photoelectrode exhibits more than

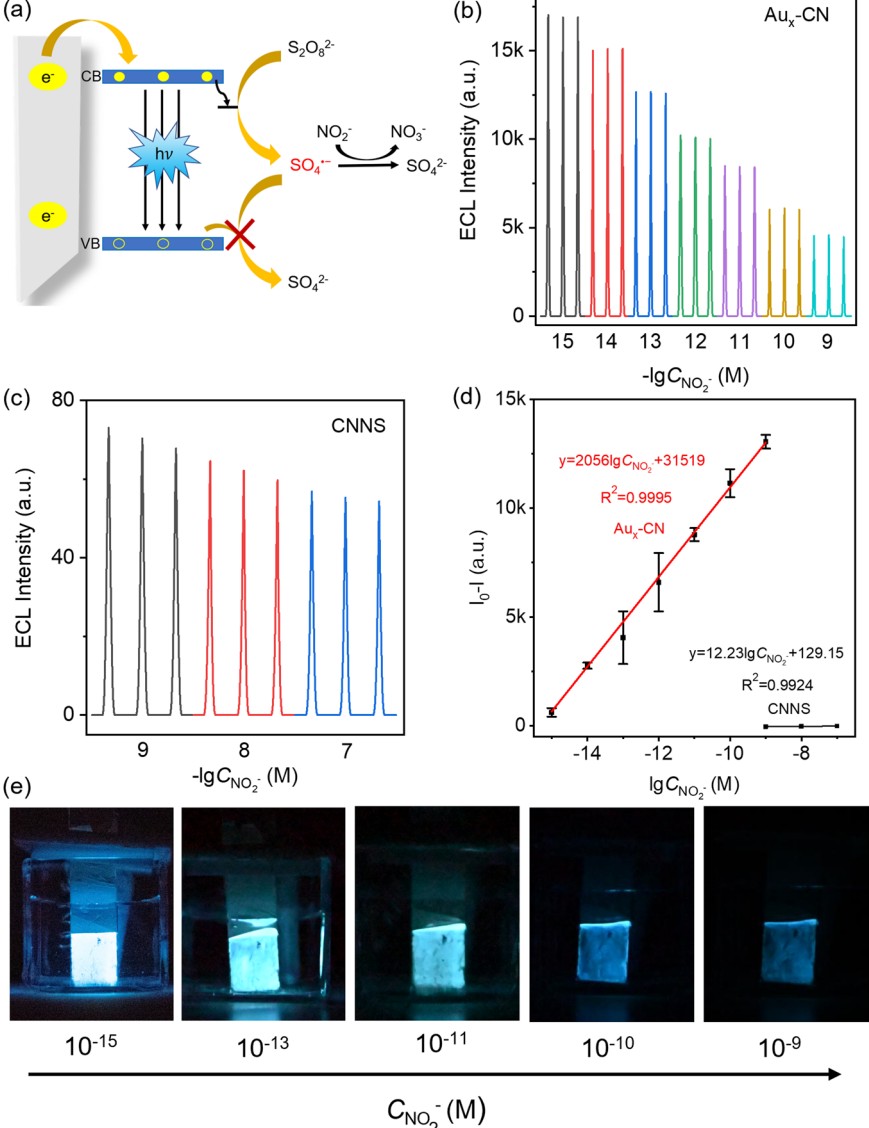

**Fig. 5 | Nitrite sensor using CN and Au$_x$-CN. a** ECL quenching mechanism upon NO$_2^-$. ECL curves in the presence of different concentrations of NO$_2^-$ at **b** Au$_x$-CN and **c** CN nanosheets photoelectrode. Different colors indicate the concentrations indicated in the x-axis. "a.u." refers to arbitrary units. **d** Calibration curve of NO$_2^-$ detection using Au$_x$-CN and CN nanosheets photoelectrode. I$_0$ and I are the ECL intensity before and after addition of NO$_2^-$, respectively. Error bars represent the standard error derived from three independent measurements. **e** Photographs of ECL at Au$_x$-CN photoelectrode in solution containing different concentrations of NO$_2^-$.

150-fold higher and 3 orders of magnitude than those of the control CN nanosheets (Fig. 5d). It suggested that the emitters with higher $\Phi_{ECL}$ held great potential in developing chemical sensors with superior sensitivity. Moreover, because of the exceptionally high cathodic ECL efficiency, it could be further developed into a visual cathodic ECL biosensor by naked eyes with uncompromising performance (Fig. 5e).

To explore the impact of doping other metals into CN on its ECL signal, the Ag$_x$-CN photoelectrode was synthesized by similar methods and conditions as that of Au$_x$-CN except that HAuCl$_4$·4H$_2$O was replaced by AgNO$_3$. As shown in Supplementary Fig. 37, the ECL intensity of Ag$_x$-CN photoelectrode exhibits almost 5 times improvement compared to that of the undoped CN photoelectrode. This enhancement in ECL intensity may be attributed to the recombination of the $d$ orbitals of the metals with the $2p$ orbitals of N atoms. The $5d$ orbitals of Au(I) were directly involved in the composition of the VBM (Fig. 4b), as another IB group metal Ag(I), the situation may be similar with Au(I).

## Discussion

In summary, we report that the timescale coordination strategy remarkably improved the performance of ECL. The Au$_x$-CN photoelectrode demonstrated a four-fold enhancement of $\Phi_{ECL}$ for CN, setting a new cathodic $\Phi_{ECL}$ record of carbon nitrides in aqueous solution and co-reagent pathway (more than 3000 times that of the Ru(bpy)$_3$Cl$_2$/K$_2$S$_2$O$_8$ reference). Notably, this exceptionally high ECL performance ranks second only to core/shell II-VI quantum dots but the latter have critical biocompatibility and environmental concerns due to heavy metals. Operando EIS studies revealed that the Au$_x$-CN photoelectrode developed 2 times shorter carrier diffusion lifetime in microsecond timescale as compared to the CN photoelectrode. Furthermore, TAS revealed that, as compared to the CN photoelectrode, there was a large portion of electrons in shallow electron trap states in Au$_x$-CN and the lifetime of these electrons was extended by 2 times of magnitude into the picosecond timescale, which accelerated the slow S$_2$O$_8^{2-}$ reduction at the emitter/co-reactants interface with the timescale of millisecond by 4 times. Meanwhile, the average deep electron

lifetime observed for $Au_x$-CN photoelectrode was more than 2 times shorter than that of the CN photoelectrode in second timescale, which can be attributed to the faster electron transfer for $Au_x$-CN photoelectrode. Thus, the emissive electron-hole recombination rate and efficiency for $Au_x$-CN photoelectrode were promoted in nanosecond timescale.

As such, operando EIS, TAS, and transit OCP collectively provided evidence that the shallow electron trap states-induced timescale coordination associated with Au-N bonds were key to the ECL characteristics of $Au_x$-CN photoelectrode. We ascribed the significant enhancement to a synergetic effect of the construction of Au-N bonds between CN layers, which provided the pathways for electron transfer to the photoelectrode/electrolyte interface. More importantly, the shallow trap state could act as an electron sink, which coordinated the timescale of the fast electron transfer in the bulk emitter and the slow redox reaction of co-reagent at diffusion layers, ultimately accelerating the recombination rate for electron-hole pairs and further promoting the ECL performance of $Au_x$-CN photoelectrode.

As a proof-of-concept application, $Au_x$-CN photoelectrode was successfully applied in a visual ECL sensor for a typical environmental contaminant, $NO_2^-$, with a wider detection range and lower detection limit, compared to the most previously studied CN nanosheets. Besides, except for the biosensing, carbon nitrides in bioimaging field could be envisaged. Due to its inherently high quantum yields, exceptional stability, excellent biocompatibility, and non-toxicity, the tunable ECL of carbon nitride holds potential for expanded applications in bioimaging[8,63,64]. The functionalization of carbon nitrides with diverse functional groups, which impart selective targeting capabilities, could facilitate the intracellular sensing of various analytes. The complete quantitative description of ECL kinetics and harnessing shallow electron trap states in timescale coordination of each step would expand the applicability of ECL emitters in various fields of optoelectronics devices, clinical diagnosis and bioimaging.

## Methods

### Reagents

Cyanamide (99%), and chloroauric acid ($HAuCl_4\cdot4H_2O$) were purchased from Energy Chemical, China. Potassium peroxodisulfate ($K_2S_2O_8$), potassium chloride (KCl), sodium dihydrogen phosphate dihydrate ($NaH_2PO_4\cdot2H_2O$), disodium hydrogen phosphate dodecahydrate ($Na_2HPO_4\cdot12H_2O$), sodium nitrite ($NaNO_2$), sodium borohydride ($NaBH_4$), methanol and triethanolamine (TEOA) were obtained from Shanghai Macklin Biochemical Co. Ltd., China. Gold nanoparticle (Au NPs, 10 nm) made by citric acid reduction method were purchased from Science Compass (China). Fluorine-doped tin oxide (FTO) glasses (12–14 Ω/sq, Zhuhai Kaivo Optoelectronic Technology Co., Ltd., China) were ultrasonically washed with acetone, ethanol, and ultrapure water for 15 min, respectively, and then dried with high purity nitrogen gas flow before use. Ultrapure water (18.2 MΩ·cm) was obtained from a Direct-Q 3 UV pure water purification system (Millipore, USA) throughout all experiments. Unless otherwise specified, all the other reagents were of analytical grade and used without further purification.

### Characterization

Fourier-transformed infrared spectra (FTIR) were recorded using a Nicolet iS10 FT-IR spectrometer, equipped with an attenuated total reflection (ATR) setup (Thermo, USA). The structure of CN and $Au_x$-CN were characterized by matrix-free laser desorption/ionization time-of-flight mass (LDI-TOF-MS, AB Sciex 5800, USA). The morphology of CN, $Au_x$-CN, and the control Au/CN photoelectrode were investigated by FEI Inspect F50 scanning electron microscope (FEI, USA). The transmission electron microscopy (TEM) and energy dispersive spectrometer (EDS) of $Au_x$-CN were investigated by JEOL JEM-2100F and Oxford Instruments X-Max. The high-angle annular dark field scanning

transmission electron microscopy (HAADF-STEM) images were performed by JEM-ARM300F GRAND ARM. The mechanical properties of the CN and $Au_x$-CN films were tested using Bruker Hysitron TI98 (Bruker, USA). The XRD patterns were measured by using Ultima IV (Rigaku, Japan). The loadings of Au were measured on an inductively coupled plasma optical emission spectrometer (ICP-OES) on an Agilent 7800 (USA). X-ray photoelectron spectroscopy (XPS) was taken on a Scientific K-Alpha electron spectrometer (Thermo, USA) with monochromatic Al Kα X-rays (hv = 1486.6 eV) as the excitation source, and the binding energy was corrected by reference C 1 $s$ level to 284.6 eV to compensate for the specimen charging. The UV-vis absorption spectra were measured on a Cary 100 (Agilent, Singapore) with a diffuse-reflectance accessory, and $BaSO_4$ was used as a standard reference (100% reflectance). The fluorescence (FL) spectra and the time-resolved FL were performed on a Fluoromax-4 (Horiba Jobin Yvon, Japan). The fs-transient absorption spectra (fs-TAS) were carried out with a commercial transient absorption spectrometer (HELIOS, Ultrafast system) that includes a 1 kHz Solstice (New Corp.). A digital camera of Nikon Z5 (Nikon, Japan) equipped with the AstrHori 35 mm F/1.8 was used to take photographs of CN and $Au_x$-CN photoelectrodes under -1.5 V in Fig. 2d. The international standards organization (ISO), aperture size, and shutter speed were set at 51200, F1.8, and 1/60, respectively. Au L3-edge analysis was performed with Si (111) crystal monochromators at the BL11B beamlines at the Shanghai Synchrotron Radiation Facility (SSRF, Shanghai, China). Before the analysis at the beamline, samples were pressed into thin sheets with 1 cm in diameter and sealed using Kapton tape film. The XAFS spectra were recorded at room temperature using a 4-channel Silicon Drift Detector (SDD) Bruker 5040. Au L3-edge extended X-ray absorption fine structure (EXAFS) spectra were recorded in transmission mode. Negligible changes in the line shape and peak position of Au L3-edge XANES spectra were observed between two scans taken for a specific sample. The XAFS spectra of these standard samples (Au foil, AuCl, and $HAuCl_4$) were recorded in transmission mode. The spectra were processed and analyzed by the software codes Athena and Artemis.

### Preparation of CN and $Au_x$-CN photoelectrode

The CN and $Au_x$-CN photoelectrode were obtained by the crystallization method. Briefly, 10 g cyanamide with or without $HAuCl_4\cdot4H_2O$ (10 μL, 500 mg/mL) was heated at 80 ℃ and melted. To deposit cyanamide crystal directly on the FTO glass, the clean FTO glass was immersed in a hot (80 °C) saturated cyanamide solution for 1 s and subsequently removed, yielding uniform crystal films on the FTO glass, which were then cooled naturally to room temperature. Finally, the crystal films were placed in a sealed glass tube and thermally condensed at 500 °C in a tube furnace (OTF-1200X-S, Hefei Kejing Materials Technology Co., Ltd, China) for 4 h in a $N_2$ atmosphere, and the as-obtained photoelectrodes were denoted as CN and $Au_x$-CN, respectively.

### Preparation of $Ag_x$-CN photoelectrode

$Ag_x$-CN photoelectrode were synthesized by the similar methods and conditions as that of $Au_x$-CN except that $HAuCl_4\cdot4H_2O$ was replaced by $AgNO_3$.

### Preparation of control Au/CN photoelectrodes

$NaBH_4$ reduction: Different concentrations of $HAuCl_4$ solution (1 nM-1 mM) was added to 8 mL of water containing the CN photoelectrode under stirring. Afterward, 48 μL of sodium citrate solution (0.01 M) was added dropwise into the suspension, followed by stirring for 30 min. Then, 120 μL of freshly prepared $NaBH_4$ solution (0.01 M) was added quickly to the above suspension and the stirring reaction was maintained for 20 min. Finally, the obtained $Au_{NaBH4}$/CN photoelectrode was immersed in ultrapure water to remove excess $NaBH_4$, sodium citrate, and unbound Au species.

Calcination: Different concentrations of HAuCl$_4$ solution (10 nM-1 mM) were dropwise to the above CN photoelectrode, then dried in a vacuum at 50 °C, followed by annealing at 300 °C in an N$_2$ atmosphere for 2 h. Finally, Au$_{Cal}$/CN was immersed in ultrapure water to remove excess unbound Au species.

Drop cast: A mixed solution of different volumes of Au NPs (2.5 µL-60 µL) and chitosan was dropwise to the CN photoelectrode, then dried in a vacuum at 50 °C. The Au$_{NPs}$/CN was immersed in ultrapure water to remove excess unbound Au NPs.

Photoreduction: A different concentration of HAuCl$_4$ solution (1 nM-1 mM) was added to 30 mL methanol containing the CN photoelectrode under stirring, followed by irradiating under a 150 W Xe light for 20 min. The Au$_{photo}$/CN was immersed in ultrapure water to remove excess unbound Au species.

## Information of carbon nitride photoelectrodes
Thickness: ca. 300 nm; area: ca. 1 cm$^2$ and mass loading: 5 mg/cm$^2$.

## ECL Measurements
The ECL intensity measurements were carried on an ECL analyzer system (MPI-E, Xi'an Ruimai Analytical Instruments Co. Ltd., China). The voltage of the photomultiplier tube (PMT) for collecting the ECL signal was biased at 100 V during detection.

Relative ECL efficiency determination: To compare ECL efficiency ($\Phi_{ECL}$) with different luminophores, a facile Ru(bpy)$_3$Cl$_2$/K$_2$S$_2$O$_8$ aqueous system was used as a reference in this study. The ECL emission spectra were recorded by integrating CHI 400C with a Fluoromax-4 FL spectrophotometer, where the slit width was 3 nm. $\Phi_{ECL}$ was defined as the ratio of the number of photons produced per electron transferred between the oxidized and reduced analyte species relative to that of Ru(bpy)$_3$Cl$_2$/K$_2$S$_2$O$_8$, using Eq. (7)[65,66]:

$$\phi_{ECL} = \frac{\left(\frac{\int ECL\,dt}{\int Current\,dt}\right)_x}{\left(\frac{\int ECL\,dt}{\int Current\,dt}\right)_{st}} \times 100\% \qquad (7)$$

where "ECL" and "Current" represent integrated ECL photon numbers from the corrected ECL spectrum according to the count sensitivity of PMT at different light wavelengths and Faradaic electrochemical current values, respectively, "st" refers to the Ru(bpy)$_3$Cl$_2$/K$_2$S$_2$O$_8$ standard and "x" refers to the analyte. The potential was fixed at −1.5 V vs. Ag/AgCl by chronoamperometry in 0.01 M phosphate buffer saline (pH 7.4) containing 25 mM K$_2$S$_2$O$_8$ and 0.1 M KCl.

Calculation of photon counts: In this work, the spectrofluorometer coupled potentiostat was used as a high-resolution ECL spectrum acquisition system. As known, the recorded emission spectrum would be distorted by the response function of the PMT (sensitivity as a function of wavelength). In this sense, the variability in PMT's sensitivity to ECL emission at different wavelengths should be calibrated. In addition, the distance from the Au$_x$-CN photoelectrode or/and GCE surface to the PMT surface and the Au$_x$-CN photoelectrode or/and GCE surface area were the same when collecting photons from the Au$_x$-CN photoelectrode and Ru(bpy)$_3$$^{2+}$.

Calculation of electrons: Unlike the Faradaic current, the non-Faradaic current during an electrochemistry process does not contribute to the ECL generation and should be subtracted when determining the intrinsic $\Phi_{ECL}$. In this work, the potential was fixed at −1.5 V vs. Ag/AgCl by chronoamperometry instead of the CV curve when collecting the ECL emission spectrum. The charge consumed by Faraday processes, including K$_2$S$_2$O$_8$ and CN reduction in ECL, can be quantitatively evaluated by subtracting the charges consumed in electrolyte without K$_2$S$_2$O$_8$ from that with K$_2$S$_2$O$_8$. It was because the reduction of K$_2$S$_2$O$_8$ was performed on CN, which originally accepted electrons from the FTO substrate electrode. In the absence of K$_2$S$_2$O$_8$ in the electrolytes, only a

minor non-Faraday charging current of CN was observed (Fig. 2a). Such non-Faraday charging also exist during the reduction of K$_2$S$_2$O$_8$, and thus should be subtracted from the total consumed electrons. Lastly, at the beginning of the i-t curve for ECL reaction, the current drops rapidly within a few seconds, corresponding to the charging current. It does not contribute to the ECL generation. Therefore, the electron should be calculated after the i-t curve reaches a plateau.

As discussed in the reports by Ding and co-workers[66], the general utilization of 5% Ru(bpy)$_3$$^{2+}$ efficiencies that are not in acetonitrile, not with a rotating ring-disk electrode, not as the same concentration, or in co-reactant systems, has created poor comparisons to measured results for almost 4 decades. The measurement of absolute ECL efficiency (number of generated photons per occupied electrons) is the ultimate solution but requires sophisticated homemade instruments and future popularization.

In this sense, using a facile Ru(bpy)$_3$Cl$_2$/K$_2$S$_2$O$_8$ aqueous system under the same conditions as the reference would be a practical way to compare the relative ECL efficiency among different aqueous ECL systems. This method is also proposed by Ding and co-workers[65]. Nonetheless, it should be noted that the ECL intensity that is often measured by photomultiplier should be corrected when it is applied in Eq. (7), as the count sensitivity of photomultiplier varies significantly to lights at different wavelengths. Many previous reports ignored this key point. We adopted such a correction in this work, making the comparison of relative ECL efficiency for different carbon nitrides more reliable and reasonable[34,35,42].

## In-situ electrochemical impedance spectroscopy (EIS) measurements
The in-situ electrochemical impedance spectroscopy (EIS) measurements were recorded in a Reference 600 potentiostat/galvanostat/ZRA (Gamry, USA). The potentials were measured against the Ag/AgCl (saturated KCl). EIS experiments were performed in a typical three-electrode system, consisting of CN or Au$_x$-CN photoelectrode, platinum wires, Ag/AgCl in saturated KCl as the working electrode, counter electrode, and reference electrode, respectively. EIS tests were performed using a 10 mV amplitude at different applied potentials versus Ag/AgCl in the frequency range of 0.1–100000 Hz. Electrolytes: 0.01 M phosphate buffer saline, 0.1 M KCl, and 25 mM K$_2$S$_2$O$_8$. In the simplified equivalent circuit model (Fig. 3d inset), R$_t$ represented the resistivity of electron transport in the emitter film, which was the intrinsic resistance of the electrode materials. R$_{ct}$ was the charge reaction resistance at the ECL emitter/S$_2$O$_8$$^{2-}$ interface, which was caused by the Faradaic reaction. The constant phase element (CPE) represented non-ideal capacitance, which was associated with the C$_n$. R$_s$ was solution resistance. The suppressed semi-circle in high frequencies and straight line in low frequencies (<45°) implies that our system deviates from the most conventional model. The non-homogeneity and roughness structure of CN photoelectrode would influence the double-layer capacitance and semi-infinite diffusion of S$_2$O$_8$$^{2-}$ ions, respectively. To compensate for these non-ideal situations, the capacitor and Warburg element in the typical Randles circuit were replaced by two constant phase elements (CPE1 and CPE2)[67], as shown in Fig. 3d inset.

## Photoelectrochemical measurements
All the electrochemical measurements were performed with a conventional three-electrode system, consisting of CN or Au$_x$-CN photoelectrode, platinum wires, Ag/AgCl in saturated KCl as the working electrode, counter electrode, and reference electrode, respectively. The photoelectrochemical (PEC) experiments were measured out in 0.1 M KCl at ambient conditions. The light source to simulate the sunlight was obtained from a 150 W Xe lamp and the average light intensity was 100 mW/cm$^2$. The open circus potential (OCP) was recorded in a Reference 600 potentiostat/galvanostat/ZRA (Gamry, USA). The potentials were measured against the Ag/AgCl (saturated KCl).

## Computational methods

All theoretical calculations were performed based on DFT, implemented in the Vienna ab initio simulation package[68,69]. For the simulation of Au incorporated in the bulk phase of CN, a $1 \times 1 \times 2$ supercell of pristine bulk CN was adopted. And the k-points were sampled in a $3 \times 3 \times 2$ Monkhorst-Pack grid. The electron exchange and correlation energy were treated within the generalized gradient approximation in the Perdew-Burke-Ernzerh of functional[70,71]. The valence orbitals were described by plane-wave basis sets with cut-off energies of 500 eV. The atomic coordinates were fully relaxed using the conjugate gradient method[72]. The convergence criteria for the electronic self-consistent iteration was set to $10^{-5}$ eV. To quantitatively compare the degree of charge transfer, a Bader charge analysis has been carried out[73]. Electronic-structure calculations were performed using the hybrid functional HSE06 to avoid underestimating the band gap by the pure DFT method[74,75]. The electronic structures and density of states were generated using the VASPKIT script[76]. The crystal structures were visualized using VESTA software[77].

## Finite element analysis

The simulations were performed using COMSOL Multiphysics (COMSOL Burlington, MA.). The transport of the diluted species (tds) module was used to simulate the electrochemical reaction that occurred on the electrodes is given by

$$S_2O_8^{2-} + 2e^- \rightleftharpoons 2SO_4^{2-} \tag{8}$$

Diffusion equations for $S_2O_8^{2-}$ and $SO_4^{2-}$ are given by

$$\frac{\partial c_i}{\partial t} = D_i \left( \frac{\partial^2 c_i}{\partial x^2} \right) \tag{9}$$

where the diffusion coefficients of $S_2O_8^{2-}$ and $SO_4^{2-}$, $D_i$ are taken as $1 \times 10^{-5}$ cm$^2$/s. The electron transfer rate of the reaction $v_{et}$ is given by the Butler-Volmer model as

$$v_{et} = k_{red} c_{S_2O_8^{2-}} - k_{ox} c_{SO_4^-} \tag{10}$$

$$k_{red} = k_0 \exp\left[-\alpha F\left(E - E^{0'}\right)/RT\right] \tag{11}$$

$$k_{ox} = k_0 \exp\left[(1-\alpha)F\left(E - E^{0'}\right)/RT\right] \tag{12}$$

where $k_{red}$ and $k_{ox}$ are the reduction and oxidation rate constants, $k_0$ is the standard electron-transfer rate constant, $\alpha$ is the transfer coefficient, F is Faraday's constant, E is the electrode potential, $E^{0'}$ is the formal potential of the redox couple, R is the gas constant, and T is the temperature. For this reaction, $\alpha$ is taken as 0.5. For the simulation of the LSVs in this study, since the LSV occurs at negative potentials and the initial solution does not contain, Eq. 10 can be approximated as

$$k_{et} = k_0 \exp\left[-\frac{\alpha F\left(E - E^{0'}\right)}{RT}\right] c_{S_2O_8^{2-}} \tag{13}$$

which is further rearranged as

$$k_{et} = k_0 \exp\left[\frac{\alpha F E^{0'}}{RT}\right] \times \exp\left[-\frac{\alpha F E}{RT}\right] c_{S_2O_8^{2-}} = A \exp\left[-\frac{\alpha F E}{RT}\right] c_{S_2O_8^{2-}} \tag{14}$$

where $A = k_0 \exp\left[\frac{\alpha F E^{0'}}{RT}\right]$. Finally, a current response for LSV, i, is given by

$$i = nFS v_{et} \tag{15}$$

where n is the number of electrons transfer, and S is the surface area of electrodes.

## Reporting summary

Further information on research design is available in the Nature Portfolio Reporting Summary linked to this article.

## Data availability

The data supporting the conclusions of this study are present in the paper and the Supplementary Information. The raw data sets used for the presented analysis within the current study are available from the corresponding authors upon request. Source data are provided with this paper.

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

## Acknowledgements

We acknowledge Prof. Jinquan Chen and Dr. Menghui Jia at the Materials Characterization Center of East China Normal University Multifunctional Platform for Innovation for their assistance in TAS measurements. We also thank Prof. Ran Chen at the School of Chemistry and Chemical Engineering, Southeast University, for his assistance in finite element analysis. This work was supported by the National Natural Science Foundation of China (22174014 and 22074015).

## Author contributions

Y. Z. and Y. F. conceived and designed the experiments. Y. F. performed the synthesis, characterization, activity evaluation, mechanism studies, and sensor of CN and Au$_x$-CN photoelectrode. H. Y. performed the DFT calculation. Y. H. and W. L. assisted in the preparation of photoelectrodes. All authors contributed to the analysis and discussion of the results. Y. F., H. Y., and Y. Z. co-wrote the manuscript, and Y. Z., S. L., and Y. S. revised the manuscript. All authors reviewed the manuscript. Y. Z. supervised the project.

## Competing interests

The authors declare no competing interests.
