## [Peer Review File · Nature Communications]

Timescale Correlation of Shallow Trap States Increases Electrochemiluminescence Efficiency in Carbon NitridesReviewer #1 (Remarks to the Author):

The slow and varied electron transfer processes limit the efficiency of ECL in bioassays and other applications. This work reports carbon nitride materials with "shallow electron trap states" using gold-nitrogen bonds, effectively syncing up the different timescales involved in ECL. Interestingly, this strategy dramatically improved ECL efficiency by setting a record for carbon nitrides. In addition, it enables a highly sensitive ECL sensor to detect nitrite ions in the environment. The authors studied the ECL properties and physicochemical insight carefully with several advanced spectroscopy, and the experimental results were fully supported by the theoretical calculations. Overall, I think this manuscript is well-written and of high novelty. However, there are still some important issues to be addressed.

- (1) Is it possible that the presence of Au species increases the ECL intensity by increasing the adsorption of $S_2O_8^{2-}$? Such factors should also be taken into consideration.
- (2) As shown in Figures 2d and 5e, the luminous intensity of the CN and Aux-CN photoelectrode is not uniform. Does the uneven area affect the flatness and luminous intensity of the photoelectrode?
- (3) Process (i), (ii), and (iii) present in Figure 3a are not mentioned in the main text. More discussion is necessary and should be associated with Eq. 1-4.
- (4) What is the meaning of "CNNS" in the caption of Figure 5d?
- (5) Since stability is essential for a typical bioassay, the authors should provide this data.

Reviewer #2 (Remarks to the Author):

Zhang et al., reports the fabrication, characterization, ECL properties, and sensing applications of Au-functionalized carbon nitrides with significantly enhanced ECL efficiency. The key innovation lies in engineering "shallow electron trap states" within the CN structure and the extended lifetime of shallow trap electrons effectively bridges the gap between the fast bulk electron transfer and the slow co-reactant reduction at the interface. As a result, a record cathodic electrochemiluminescence efficiency of carbon nitrides was obtained. This work makes a significant advancement in electrochemiluminescence (ECL) efficiency using carbon nitrides and the concept of timescale coordination would be useful for other ECL luminophores. Nevertheless, the manuscript in the present state needs several improvements before it can be considered for publication in Nature Communications.

1. There are several terminal groups (such as $-NH_2$ and $C\equiv H$) on the surface of carbon nitrides. Can the type and quantity of these functional groups increase ECL intensity after the introduction of Au species?
2. Both Au nanoparticles and single atoms were observed in the as-prepared Aux-CN. For Au-N functionalization, more discussion is needed to explain why and how they cooperate and contribute to the creation of an electron shallow trapping state.
3. Several typos, inconsistencies, and styles require revision. For instance, comparative literatures on the ECL efficiency of other emitters are lacking in Figure 2f and there is a misspelling of "Aux-/CN" in the caption; the electron transfer lifetime for the shallow state is abbreviated as τ_{avg} -shallow in the table, while as τ_{ave} -shallow in the main text; both Eq. and Eqn. is used to denote Equation; for clarity, in Figure 3a, the CB and VB positions should be plotted in the scheme; the words in Fig. 3 are too small and difficult to read.

Reviewer #3 (Remarks to the Author):

Fang et al reported an unusually enhanced electrochemiluminescence efficiency of carbon nitrides via a timescale coordination strategy through Au-N bond functionalization. Both experimental studies and theoretical calculations demonstrated that Au-N bonds endowed shallow trapped electron states, which coordinated the timescale of fast electron transfer in the bulk emitter and slow redox reaction of the co-reagent at diffusion layers. In general, the transformation of effort from length to time scale is very interesting and can be regarded as a significant advance in

insight into boosting the ECL efficiency of emitters. Therefore, I would like to suggest the following minor revisions before further consideration for publication:

1. Figure 1a shows that the samples were annealed at 500°C for 2 hours for the preparation of the film on FTO. However, the authors mentioned that the samples were annealed for 4 hours on Page 2 of the manuscript. What are the correct descriptions?
2. The authors report the Au can enhance the ECL intensity. Can other metals (e.g., Ag, Pt, and Pd) play the same role? More explanations are needed.
3. Electroluminescent emitters with high efficiency are also highly envisioned for bioimaging applications. A more comprehensive discussion of this concern would be helpful for a broader readership.
4. What about the stability of the Au-CN? This is vital for practical applications.

Reviewer #4 (Remarks to the Author):

Fang et al. report the electrogenerated chemiluminescence (ECL) of carbon nitride (CN) films incorporating Au nanoparticles. The electrochemical and ECL properties including DFT calculations of such Au-CN photoelectrode are then described. Finally, they present an analytical application with the detection of nitrite ion. The authors claimed a huge ECL efficiency that is not fully supported by the data and by the comparison table. In addition, they claimed that they used "a timescale coordination strategy to improve Φ_{ECL} of carbon nitrides". This point is not demonstrated. Considering the major points listed below, the reviewer does not recommend the publication of the manuscript in Nature Communications.

Major points:

- STEM images of the Au-CN film show a wide dispersion of Au nanoclusters in the CN. It indicates that the control of the fabrication process is not well-controlled and therefore it makes difficult the rationalization of the properties.
- The cyclic voltammograms (Figure 2) show a very poor characterization of the Au-CN material. One cannot observe any clear reduction wave.
- Ding's group reported the absolute determination of the ECL efficiency (J. Phys. Chem. C 2021, 125, 22274). The authors should recalculate the ECL efficiency by considering the rigorous Ding's method.
- On page 8, the authors describe the electron-transfer processes in Au-CN film and the ECL emission. ECL is generated in reduction with the K₂S₂O₈ co-reactant. However, they compare it with the tripropylamine co-reactant (Figure S13) that generates ECL only in oxidation. This point should be corrected.

Point-by-point response to comments for Manuscript: NCOMMS-24-01432

Reviewer 1:

Q: The slow and varied electron transfer processes limit the efficiency of ECL in bioassays and other applications. This work reports carbon nitride materials with "shallow electron trap states" using gold-nitrogen bonds, effectively syncing up the different timescales involved in ECL. Interestingly, this strategy dramatically improved ECL efficiency by setting a record for carbon nitrides. In addition, it enables a highly sensitive ECL sensor to detect nitrite ions in the environment. The authors studied the ECL properties and physicochemical insight carefully with several advanced spectroscopy, and the experimental results were fully supported by the theoretical calculations. Overall, I think this manuscript is well-written and of high novelty. However, there are still some important issues to be addressed.

A: We thank the reviewer for the valuable suggestions. The detailed response to the comments is shown as follows.

Q1: Is it possible that the presence of Au species increases the ECL intensity by increasing the adsorption of $S_2O_8^{2-}$? Such factors should also be taken into consideration.

A1: Good question! To examine the impact of adsorption on the ECL system, the control Au/CN photoelectrodes (Au species on the surface of the CN photoelectrode) were synthesized using the most common methods, including calcination, drop casting, sodium borohydride ($NaBH_4$) reduction, and photoreduction. As shown in **Figure S29**, the ECL emission intensity of the Au/CN photoelectrodes, varying in Au content, did not demonstrate evident enhancement. It indicated that the Au species did not exhibit a noticeable adsorption effect on the $S_2O_8^{2-}$ in this study.

Therefore, for clarity, the above discussion has been added to the revised Supplementary Information after **Figure S29** (Page S41).

Q2: As shown in Figures 2d and 5e, the luminous intensity of the CN and Au_x-CN photoelectrode is not uniform. Does the uneven area affect the flatness and luminous intensity of the photoelectrode?

A2: Some uneven regions of ECL on CN or Au_x-CN photoelectrodes were observable. It was presumably because of the release of ammonia during the thermal condensation. As the overall ECL intensity was measured in this study, the uneven area did not have evident negative effects. Nonetheless, if these photoelectrodes are used for bioimaging in the future, more homogeneous carbon nitride film is envisioned.

Therefore, for clarity, “the uneven luminescence on the photoelectrodes may be attributed to differences in surface flatness after the thermal condensation” has been added to the revised manuscript (Page 8).

Q3: Process (i), (ii), and (iii) present in Figure 3a are not mentioned in the main text. More discussion is necessary and should be associated with Eq. 1-4.

A3: Sorry for the confusion. Process (i), (ii), and (iii) present in **Figure 3a** have been deleted and **Eq. 1-4** have been directly marked in Figure 3a.

Q4: What is the meaning of “CNNS” in the caption of Figure 5d?

A4: “CNNS” in the caption of **Figure 5d** has been revised as “CN nanosheets”.

Q5: Since stability is essential for a typical bioassay, the authors should provide this data.

A5: Yes, the stability of the sensor was a key factor in their application. As shown in **Figure S36**, the ECL intensity of the biosensor for detection of NO₂⁻ had little change under continuous potential scanning more than 10 cycles, which indicated the good stability of the biosensor.

Therefore, for clarity, the above discussion has been added to the revised manuscript on page 18.

Figure S36. ECL intensity of the as-proposed biosensor for detection of NO_2^- under consecutive cyclic potential scanning. [NEW data]

Reviewer 2:

Q: Zhang et al., reports the fabrication, characterization, ECL properties, and sensing applications of Au-functionalized carbon nitrides with significantly enhanced ECL efficiency. The key innovation lies in engineering "shallow electron trap states" within the CN structure and the extended lifetime of shallow trap electrons effectively bridges the gap between the fast bulk electron transfer and the slow co-reactant reduction at the interface. As a result, a record cathodic electrochemiluminescence efficiency of carbon nitrides was obtained. This work makes a significant advancement in electrochemiluminescence (ECL) efficiency using carbon nitrides and the concept of timescale coordination would be useful for other ECL luminophores. Nevertheless, the manuscript in the present state needs several improvements before it can be considered for publication in Nature Communications.

A: We thank the reviewer for the valuable suggestions, which we could make a substantial improvement to this manuscript. The detailed response to the comments is shown as follows.

Q1: There are several terminal groups (such as $-\text{NH}_2$ and $\text{C}\equiv\text{H}$) on the surface of carbon nitrides. Can the type and quantity of these functional groups increase ECL intensity after the introduction of Au species?

A1: Good suggestions! As shown in **Figure S4 (updated)**, both CN and Au_x-CN photoelectrodes showed the characteristic vibrations peaks for $\nu(-\text{NH}_2)$ at 3566/3462 cm^{-1} and $\text{C}\equiv\text{N}$ at 2150 cm^{-1} , respectively. Therefore, the type of the terminal groups (such as $-\text{NH}_2$ and $\text{C}\equiv\text{H}$) on the surface of carbon nitride remained unchanged after the introduction of Au species.

In addition, to more accurate comparison of the peak areas of specific functional groups across different samples, the peaks representing the triazine or heptazine ring out of plane bending at 825 cm^{-1} were normalized. The peak area of $\nu(-\text{NH}_2)$ at 3566 cm^{-1} /3462 cm^{-1} for CN and Au_x-CN photoelectrodes were 12727/7563 and 12734/6709, and the peak area of $\nu(-\text{C}\equiv\text{N}-)$ at 2150 cm^{-1} for CN and Au_x-CN photoelectrodes were 9289 and 9008, respectively. Based on these quantitative analyses, the quantity of the terminal groups (such as $-\text{NH}_2$ and $-\text{C}\equiv\text{H}-$) on the surface of carbon nitride was almost unchanged after the introduction of Au species.

Therefore, for clarity, the above discussion has been added to the revised Supplementary information after **Figure S4** (Page S16).

Q2: Both Au nanoparticles and single atoms were observed in the as-prepared Au_x-CN. For Au-N functionalization, more discussion is needed to explain why and how they cooperate and contribute to the creation of an electron shallow trapping state.

A2: Good questions! Nanoparticles and single atoms usually show different electronic structures in materials. Importantly, this discrepancy allows the electron-deficient single atoms and the electron-rich nanoparticles, hence display diverse behaviors and cooperatively promote ECL performance. In addition, the charge transfer between nanoparticles and single atoms may modulate the electronic structure of emitters, resulting in enhanced ECL performance. Within our system, by DFT calculation (**Figure 33a** and **Figure 34**), both Au nanoparticles and single atoms could form Au-N bonds between the Au atoms and the two adjacent CN layers. Quantitative ECL kinetics measurements and theoretic calculations jointly disclosed Au-N bonds endowed shallow trapped electron states, which coordinated the timescale of the fast electron transfer in the bulk emitter and the slow redox reaction of co-reagent at diffusion layers.

Therefore, for clarity, *“The Au nanoparticles and single atoms may be enabled to modulate the electronic structure of emitters, resulting in enhanced ECL performance. Moreover, by DFT*

calculation (**Figure 33a** and **Figure 34**), both Au nanoparticles and single atoms could form Au-N bonds between the Au atoms and the two adjacent CN layers. Quantitative ECL kinetics measurements and theoretic calculations jointly disclosed Au-N bonds endowed shallow trapped electron states, which coordinated the timescale of the fast electron transfer in the bulk emitter and the slow redox reaction of co-reagent at diffusion layers.” have been added to the revised Supplementary information after **Figure S34** (Page S46).

Q3: Several typos, inconsistencies, and styles require revision. For instance, comparative literatures on the ECL efficiency of other emitters are lacking in Figure 2f and there is a misspelling of “Aux-/CN” in the caption; the electron transfer lifetime for the shallow state is abbreviated as $\tau_{\text{avg-shallow}}$ in the table, while as $\tau_{\text{ave-shallow}}$ in the main text; both Eq. and Eqn. is used to denote Equation; for clarity, in Figure 3a, the CB and VB positions should be plotted in the scheme; the words in Fig. 3 are too small and difficult to read.

A3: We are sorry for the typos. We have double-checked the manuscript and all the typos were revised. The detailed changes are listed as follows:

- (1) The comparative literature on the ECL efficiency of other emitters in **Figure 2f** has been added in the inset (Ref. 34, 35 ,42, S29, S37, S44, S49).
- (2) The “Au_x-/CN” in the caption of **Figure 2f** has been revised as “Au_x-CN”;
- (3) The “**Eqn. S1**” and “**Eqn. 6**” in Supplementary information have been revised as “**Eq. S1**” and “**Eq. 6**”;
- (4) The “ $\tau_{\text{avg-shallow}}$ ” in the table has been revised as “ $\tau_{\text{ave-shallow}}$ ”;
- (5) The CB and VB positions have been plotted in the revised **Figure 3a**;
- (6) The words in **Figure 3** have been all magnified or in bold.

Reviewer 3:

Q: Fang at al reported an unusually enhanced electrochemiluminescence efficiency of carbon nitrides via a timescale coordination strategy through Au-N bond functionalization. Both experimental studies and theoretical calculations demonstrated that Au-N bonds endowed shallow trapped electron states, which coordinated the timescale of fast electron transfer in the bulk emitter

and slow redox reaction of the co-reagent at diffusion layers. In general, the transformation of effort from length to time scale is very interesting and can be regarded as a significant advance in insight into boosting the ECL efficiency of emitters. Therefore, I would like to suggest the following minor revisions before further consideration for publication:

A: We thank the reviewer for the valuable suggestions, which we could make a substantial improvement to this manuscript. The detailed response to the comments is shown as follows.

Q1: Figure 1a shows that the samples were annealed at 500 °C for 2 hours for the preparation of the film on FTO. However, the authors mentioned that the samples were annealed for 4 hours on Page 2 of the manuscript. What are the correct descriptions?

A1: Sorry for the confusion, the “2 hours” in **Figure 1a** has been revised as “4 h”.

Q2: The authors report the Au can enhance the ECL intensity. Can other metals (e.g., Ag, Pt, and Pd) play the same role? More explanations are needed.

A2: To explore the impact of doping other metals into CN on its ECL signal, the Ag_x-CN photoelectrode were synthesized by similar methods and conditions as that of Au_x-CN except that HAuCl₄·4H₂O was replaced by AgNO₃. As shown in **Figure S37**, the ECL intensity of Ag_x-CN photoelectrode exhibits almost 5 times improvement compared to that of the undoped CN photoelectrode. This enhancement in ECL intensity may be attributed to the recombination of the d orbitals of the metals with the 2p orbitals of N atoms. The 5d orbitals of Au(I) were directly involved in the composition of the VBM (**Figure 4b**), as another IB group metal Ag(I), the situation may be similar with Au(I). Nonetheless, to further understand the enhancement mechanism of ECL intensity of CN by metal doping, we will carry out further studies in the future.

Therefore, for clarity, the preparation of Ag_x-CN photoelectrode has been added in the Experimental section of the revised Supplementary information (Page S7). And the above discussion has been added to the revised manuscript (Page 18).

Figure S37. Ratio of ECL intensity for Ag_x-CN to CN photoelectrode. [NEW data]

Q3: Electroluminescent emitters with high efficiency are also highly envisioned for bioimaging applications. A more comprehensive discussion of this concern would be helpful for a broader readership.

A3: Suggestive comment! ECL emitters with high ECL efficiency (Φ_{ECL}) are of great significance in improving the contrast and sensitivity in the application of bioimaging. Carbon nitride nanofilms with inherent high quantum yields, high stability, good biocompatibility, and nontoxicity are promising candidates for bioimaging. Furthermore, carbon nitride can also be functionalized with various functional groups, which endows carbon nitrides with selective targeting properties (e.g., Ref. 8, 63, and 64: *J. Am. Chem. Soc.* **2021**, *143*, 17910; *J. Am. Chem. Soc.* 2019, *141*, 10294; *J. Am. Chem. Soc.* **2018**, *140*, 15904). Thus, further applications of carbon nitrides in the bioimaging field could be envisaged.

For clarity, the discussion “*Besides, except for the biosensing, carbon nitrides on bioimaging field could be envisaged. Due to its inherently high quantum yields, exceptional stability, excellent biocompatibility, and non-toxicity, the tunable ECL of carbon nitride holds potential for expanded applications in bioimaging.*”^{8,63,64} *The functionalization of carbon nitrides with diverse functional groups, which impart selective targeting capabilities, could facilitate the intracellular sensing of various analytes.*” along with the supporting references 8, 63, and 64 have been added in the revised manuscript (Page 19 and 20).

Q4: What about the stability of the Au_x-CN? This is vital for practical applications.

A4: The ECL of Au_x-CN photoelectrode was stable under continuous CV scans (**Figure 2c**). Moreover, the stability of the sensor was also a key factor in their application. As shown in **Figure S36**, the ECL intensity of the biosensor for detection of NO₂⁻ had little changes under continuous potential scanning, which indicated the good stability of the biosensor.

Therefore, for clarity, the above discussion has been added to the revised manuscript on page 18.

Figure S36. ECL intensity of the as-proposed biosensor for detection of NO₂⁻ under consecutive cyclic potential scanning. [NEW data]

Reviewer 4:

Q: Fang et al. report the electrogenerated chemiluminescence (ECL) of carbon nitride (CN) films incorporating Au nanoparticles. The electrochemical and ECL properties including DFT calculations of such Au_x-CN photoelectrode are then described. Finally, they present an analytical application with the detection of nitrite ion. The authors claimed a huge ECL efficiency that is not fully supported by the data and by the comparison table. In addition, they claimed that they used “a timescale coordination strategy to improve Φ_{ECL} of carbon nitrides”. This point is not demonstrated. Considering the major points listed below, the reviewer does not recommend the publication of the manuscript in Nature Communications.

A: We thank the reviewer for the valuable suggestions, which we could make a substantial improvement to this manuscript. And it also came to our great attention that we should give a

better scholarly presentation to avoid serious scientific misunderstanding of the core aspect of this manuscript. The detailed response to the comments is shown as follows.

Q1: They claimed that they used “a timescale coordination strategy to improve Φ_{ECL} of carbon nitrides”. This point is not demonstrated.

A1: As shown in **Table 1** and **Figure S28**, the shallow-trapped electrons in the bulk emitter that existed at about the picosecond time scale (τ_{shallow}) extended 2 times magnitude by the Au-N bond functionalization, which coordinated the slow charge transfer at the emitter/co-reactant interface in the millisecond time scale (τ_{d}) and fast electron transfer in bulk emitter in the microsecond time scale (τ_{n}). It would increase the reduction of co-reactants at the interface, and further improve the electron-hole recombination rate and efficiency for the ECL of $\text{Au}_x\text{-CN}$ in the nanosecond time scale (τ_{r}).

Therefore, for clarity, the above discussion has been updated in the revised manuscript (Page 14).

Figure S28. Possible mechanism for ECL of (a) CN photoelectrode and (b) $\text{Au}_x\text{-CN}$ photoelectrode with different timescale. [NEW scheme]

Q2: STEM images of the Au-CN film show a wide dispersion of Au nanoclusters in the CN. It indicates that the control of the fabrication process is not well-controlled and therefore it makes difficult the rationalization of the properties.

A2: Good questions! We have tried our best to optimize the method for synthesizing CN electrodes to be as uniform, flatness and robust as possible. In general, it seems perfect to prepare perfect single Au atom or Au nanoparticles modified CN using the well-known impregnation and pre-polymerization method. However, the poor adhesion of the modified CN powder on the substrate electrode led to a serious decrease and uncertainty in ECL efficiency evaluation and mechanism studies. For this technical challenge, our group have developed several ways to improve the quality of the photoelectrode (e.g., *ACS Appl. Mater. Interfaces* **2016**, 8, 22287), but is still unsatisfied.

Recently, we found the in-situ growth method can address this problem, and ECL efficiency gained continuous increase (*Angew. Chem. Int. Ed.* **2020**, 59, 1139; *Adv. Optical Mater.* **2022**, 10, 2201017; *Adv. Optical Mater.* **2023**, 11, 2202737). Thus, the in-situ growth method was used in this study to prepare Au_x-CN photoelectrodes. Although the size of Au species in the Au_x-CN photoelectrodes was not as controllable as that in the conventional Au_x-CN powder, the DFT calculation (**Figure 33a** and **Figure 34**) showed both Au nanoparticles and single atoms formed Au-N bonds between the Au atoms and the two adjacent CN layers, the key in introducing the shallow electron trapping states. Besides, more and more research reported that the charge transfer between nanoparticles and single atoms enables to modulation of the electronic structure of materials, resulting in enhanced performance (e.g., Ref. 23 and 24: *Applied Catalysis B: Environmental* **2022**, 307, 121193; *Energy Environ. Sci.* **2023**, 16, 5663).

Therefore, for clarity, “*It should be noted that both Au nanoparticles and single atoms were observed in the Au_x-CN photoelectrode. The DFT calculation in **Figure 33a** and **Figure 34** showed both Au nanoparticles and single atoms formed Au-N bonds between the Au atoms and the two adjacent CN layers. Quantitative ECL kinetics measurements and theoretic calculations jointly disclosed Au-N bonds endowed shallow trapped electron states, which coordinated the timescale of the fast electron transfer in the bulk emitter and the slow redox reaction of co-reagent at diffusion layers. Besides, more and more research reported that the charge transfer between nanoparticles and single atoms enables to modulation of the electronic structure of materials, resulting in enhanced performance.*^{23, 24}”, have been added to the revised Supplementary information after **Figure S34** along with the above supporting reference 23 and 24. (page S46).

Q3: The cyclic voltammograms (Figure 2) show a very poor characterization of the Au-CN material. One cannot observe any clear reduction wave.

A3: Good questions! In fact, the reduction peak of $\text{K}_2\text{S}_2\text{O}_8$ was out of the scope of the electrochemical window. It was attributed to the high iR drop of CN and $\text{Au}_x\text{-CN}$ photoelectrode. Motivated by this comment, the kinetic process of $\text{K}_2\text{S}_2\text{O}_8$ reduction was further explored by simulating the LSV curves during the ECL process using COMSOL Multiphysics.

During the simulation process, it was challenging to fit the original LSV curves of CN and $\text{Au}_x\text{-CN}$ photoelectrode (black curve in **Figure S10**) with simulation due to the different shapes of the curve around the onset potential. Considering the high resistancy of the electrode material, the iR compensation was implemented, which assumed a constant resistance R of the electrode. After iR compensation, the LSV curves of CN and $\text{Au}_x\text{-CN}$ photoelectrode could be fitted with simulation (**Figure S10**). An important point to note is that only a proper R value could lead to a good fit with simulation; too large or too small R values would lead to a heavily distorted shape of the LSV (data not shown), which is impossible to fit.

In general, such “ill”-defined CV curves were often ascribed to the nature of electrode materials. First of all, the control CV/ECL curves without $\text{K}_2\text{S}_2\text{O}_8$ (**Figure 2a**) demonstrated a minor current, indicating negligible polarization of water during the reduction of $\text{K}_2\text{S}_2\text{O}_8$. Moreover, the finite element analysis (**Figure S10**) showed the high iR drop made the reduction wave out of the scope of the electrochemical window. In contrast, the reduction wave for $\text{K}_2\text{S}_2\text{O}_8$ could be occasionally observed by low-quality CN film electrodes owing to pinholes, but which had a much lower Φ_{ECL} . It evidently suggested the transferring of the electron to $\text{K}_2\text{S}_2\text{O}_8$ bridged by the CN film could make more use of electrons in ECL, compared to the direct accessing of electrons from the substrate electrode that was independent of the follow-up ECL reactions and generally superfluous.

Therefore, to avoid this confusion, the iR drop correction using the finite element analysis has been undertaken and added in **Figure S10**, along with the above complementary discussion, in the revised Supplementary information after **Figure S10** (Page S22).

Figure S10. LSV curve simulated of (a) CN photoelectrode with $R= 395$ ohm and (b) Au_x -CN photoelectrode with $R= 195$ ohm. [New data].

Q4: Ding's group reported the absolute determination of the ECL efficiency (J. Phys. Chem. C 2021, 125, 22274). The authors should recalculate the ECL efficiency by considering the rigorous Ding's method.

A4: Insightful comments! As discussed in the above-mentioned reports by Ding and co-workers (Ding et al., J. Phys. Chem. C 2021, 125, 22274; *Anal. Chem.* **2021**, 93, 11626; Ding et al., *J. Electroanal. Chem.* **2022**, 906, 115891), the general utilization of 5% $Ru(bpy)_3^{2+}$ efficiencies that are not in acetonitrile, not with a rotating ring-disk electrode, not as the same concentration, or in co-reactant systems, has created poor comparisons to measured results for almost 4 decades. The measurement of absolute ECL efficiency (number of generated photons per occupied electrons) is an ultimate solution but requires sophisticated homemade instruments and future popularization, and currently, to the best of our knowledge, ONLY Ding's group in the world can do it.

In this sense, using a facile $Ru(bpy)_3Cl_2/K_2S_2O_8$ aqueous system under the same conditions as the reference would be a practical way to compare the relative ECL efficiency among different aqueous ECL systems. This method is also proposed by Ding and co-workers (Ding et al., *Carbon* **2018**, 129, 45). Nonetheless, it should be noted that the ECL intensity that is often measured by photomultiplier should be corrected when it is applied in **Eq. S1**, as the count sensitivity of photomultiplier varies significantly to lights at different wavelengths. Many previous reports ignored this key point. We adopted such a correction in this work, making the comparison of

relative ECL efficiency for different carbon nitride more reliable and reasonable. More discussion on relative ECL efficiency calculation and rationality is as follows:

To compare ECL efficiency (Φ_{ECL}) with different luminophores, a facile Ru(bpy)₃Cl₂/K₂S₂O₈ aqueous system was used as a reference in this study. The ECL emission spectra were recorded by integrating CHI 400C with a Fluoromax-4 FL spectrophotometer, where the slit width was 3 nm. Φ_{ECL} was defined as the ratio of the number of photons produced per electron transferred between the oxidized and reduced analyte species relative to that of Ru(bpy)₃Cl₂/K₂S₂O₈, using **Eq. S1**;

$$\phi_{ECL} = \frac{\left(\frac{\int ECL dt}{\int Current dt}\right)_x}{\left(\frac{\int ECL dt}{\int Current dt}\right)_{st}} \times 100\% \quad [\text{Eq. S1}]$$

where "ECL" and "Current" represent integrated ECL photon numbers from the corrected ECL spectrum according to the count sensitivity of PMT at different light wavelengths and Faradaic electrochemical current values, respectively, "st" refers to the Ru(bpy)₃Cl₂/K₂S₂O₈ standard and "x" refers to the analyte. The potential was fixed at -1.5 V vs. Ag/AgCl by chronoamperometry in 0.01 M PBS (pH 7.4) containing 25 mM K₂S₂O₈ and 0.1 M KCl.

Calculation of photon counts: In this work, the spectrofluorometer coupled potentiostat was used as a high-resolution ECL spectrum acquisition system. As known, the recorded emission spectrum would be distorted by the response function of the PMT (sensitivity as a function of wavelength). In this sense, the variability in PMT's sensitivity to ECL emission at different wavelengths should be calibrated. In addition, the distance from the Au_x-CN photoelectrode or/and GCE surface to the PMT surface and the Au_x-CN photoelectrode or/and GCE surface area were the same when collecting photons from the Au_x-CN photoelectrode and Ru(bpy)₃²⁺.

Calculation of electrons: Unlike the Faradaic current, the non-Faradaic current during an electrochemistry process does not contribute to the ECL generation and needs to be subtracted when determining the intrinsic Φ_{ECL} . In this work, the potential was fixed at -1.5 V vs. Ag/AgCl by chronoamperometry instead of the CV curve when collecting the ECL emission spectrum. The charge consumed by Faraday processes including K₂S₂O₈ and CN reduction in ECL could be quantitatively evaluated by subtracting the consumed charge during chronoamperometric measurements in electrolyte without K₂S₂O₈ from that with K₂S₂O₈. It was because the reduction of K₂S₂O₈ was performed on CN, which originally accepted electrons from the FTO substrate electrode. In the absence of K₂S₂O₈ in the electrolytes, only a minor non-Faraday charging current

of CN was observed (**Figure 2a**). Such non-Faraday charging should also exist during the reduction of $K_2S_2O_8$, and thus could be subtracted from the total consumed number of electrons. Lastly, at the beginning of the i-t curve for the ECL reaction, the current drops rapidly within a few seconds, corresponding to the charging current. It does not contribute to the ECL generation. Therefore, the electron should be calculated after the i-t curve reaches a plateau.

Therefore, for a better scholarly presentation, the above discussion has been added to the revised Supplementary information on Page S8 and S9.

Q5: On page 8, the authors describe the electron-transfer processes in Au-CN film and the ECL emission. ECL is generated in reduction with the $K_2S_2O_8$ co-reactant. However, they compare it with the tripropylamine co-reactant (Figure S13) that generates ECL only in oxidation. This point should be corrected.

A5: Sorry for the confusion. For co-reactant typed ECL, based on the pathways through which oxidation-reduction reactions occur, they can be categorized into three categories: (1) the co-reactant and the emitters react simultaneously at the electrode (**Figure S14a** and **c**); (2) the co-reactants react initially, after which the emitter reacts with the intermediate of the co-reactants (**Figure S14b**); (3) the emitters react first, followed by an oxidation-reduction reaction between the co-reactant and the emitters (**Figure S14d**). To elucidate these processes, the most classic ECL system ($Ru(bpy)_3^{2+}/TPrA$) was employed in the literature by Bard and co-workers.

The electrochemical impedance spectra (EIS) of Au_x -CN and FTO photoelectrodes were measured using $[Fe(CN)_6]^{3-}/[Fe(CN)_6]^{4-}$ as the electrochemical probe (**Figure S15** and **Table S2**). The redox reaction was evidently inhibited by ca. 4 orders at Au_x -CN photoelectrode in regarding of the interfacial charge transfer resistance across the electrode/electrolyte (R_{ct}), verifying the $[Fe(CN)_6]^{3-}$ obtained electron from Au_x -CN instead of the FTO. Therefore, the Au_x -CN/ $K_2S_2O_8$ system in this study was followed by the third type of reaction pathway (**Figure S14d**), i.e., the emitters react first, followed by an oxidation-reduction reaction between the co-reactant and the emitters.

Therefore, for clarity, the above discussion has been added to the revised Supplementary information after **Figure S14** and **S15** (Page S25 and S26).

Reviewer #1 (Remarks to the Author):

The authors have revised the manuscript and it is recommended to be accepted at the current stage.

Reviewer #2 (Remarks to the Author):

The authors have revised the manuscript well. Now, I agree its publication in its present form.

Reviewer #3 (Remarks to the Author):

The authors have addressed all my comments. I recommend the publication of this work.

Reviewer #4 (Remarks to the Author):

The authors have greatly improved the quality of the manuscript by providing a comprehensive mechanism for the ECL process. In addition, they have replied to the different comments and clarified the corresponding points. Therefore, considering the original of their approach and of their results, I recommend the publication of this manuscript.